# Temperature Sensitivity of Soil Respiration in Two Temperate Forest Ecosystems: The Synthesis of a 24-Year Continuous Observation

Irina Kurganova * , Valentin Lopes de Gerenyu , Dmitry Khoroshaev , Tatiana Myakshina, Dmitry Sapronov and Vasily Zhmurin

Institute of Physicochemical and Biological Problems of Soil Sciences, Russian Academy of Sciences, 142290 Pushchino, Russia
* Correspondence: ikurg@mail.ru; Tel.: +7-917-5240342

**Abstract:** Soil respiration (SR) is one of the largest fluxes in the global carbon cycle. The temperature sensitivity of SR (often termed as $Q_{10}$) is a principal parameter for evaluating the feedback intensity between soil carbon efflux and global warming. The present study aimed to estimate the seasonal and interannual dynamics of the temperature sensitivity of SR based on a long-term 24-year series of measurements in two temperate forest ecosystems in European Russia. The study was conducted in a mature mixed forest with sandy Entic Podzol and in a secondary deciduous forest with loamy Haplic Luvisol. The SR rate was measured continuously from December 1997 to November 2021 at 7–10-day intervals using the closed chamber method. Sandy Entic Podzol demonstrated a higher temperature sensitivity of SR in comparison with loamy Entic Luvisol. The $Q_{10}$ values for both soils in dry years were 1.3–1.4 times lower than they were in the years with normal levels of humidity. For both types of soil, we observed a significant positive correlation between the $Q_{10}$ values and wetness indexes. The interannual variability of $Q_{10}$ values for SR in forest soils was 18%–40% depending on the calculation approach and levels of aridity/humidity over the growing season. The heterogeneous $Q_{10}$ values should be integrated into SR and C balance models for better predictions.

**Keywords:** $CO_2$ emission; $Q_{10}$ and $SR_{10}$ values; empirical models; meteorological indexes; long-term monitoring; interannual variability; Entic Podzol; Haplic Luvisol; mixed and deciduous forests; climate change

## 1. Introduction

Forest ecosystems play a key role in the global biogeochemical carbon (C) cycle, providing the principal part of the global carbon sink in terrestrial ecosystems [1,2]. For this reason, the forests are the main climate-stabilizing systems of the earth's surface [3]. The Russian Federation has the largest forest area in the world, amounting to ca. 20% of all forests across the globe (https://www.fao.org/forest-resources; accessed on 1 January 2020). In the Northern Hemisphere, Russia's forest ecosystems provide more than 50% of the total C sink in the forest area, contributing significantly to the regulation of the global cycle of organic C [4,5].

Soil respiration (SR) is one of the largest fluxes in the global C cycle [6–8], but until now the regional and global evaluations of SR magnitude are the most uncertain constituents of the terrestrial C cycle [9]. SR is usually defined as $CO_2$ released from the soil into the atmosphere via the activity of two main processes: decomposing litter and soil organic matter by micro- and macro-organisms (heterotrophic respiration, HR) and root (or autotrophic) respiration, RR [10–12]. SR also includes the non-biological production of $CO_2$, but this flux is usually negligible [8]. SR fluxes demonstrate high spatial [9,13,14] and temporal heterogeneity [15–17]. The most important abiotic factors influencing SR are the soil temperature and soil water availability, whereas the substrate quality and

availability are the main biotic factors, which are governed by the type and productivity of the vegetation [18,19]. Since the temporal and spatial dynamics of SR are the integrated result of a broad spectrum of biological processes, which are controlled by wildly varying environmental constraints, SR magnitudes remain difficult to model or predict [9,20].

Until now, temperature has been considered the most significant factor that influences the SR magnitude on a global scale, as well as on local and ecosystem levels [14,16,21]. The temperature sensitivity of SR is a principal parameter for evaluating the feedback intensity between soil carbon efflux and global warming [11,18,22,23]. The temperature sensitivity of SR (often termed as $Q_{10}$) is a factor by which the SR magnitude is multiplied when the temperature increases by 10 °C [19,24]. This parameter is applied in several models, which usually employ a fixed value of 1.5 or 2 for all soils and at all levels of soil moisture [9,25,26]. The mean $Q_{10}$ values for different biomes range from 1.43 to 2.03, with the highest value in tundra and the lowest value in deserts [11]. When spatially heterogeneous $Q_{10}$ values were integrated into SR models instead of a fixed $Q_{10}$ value, 40% higher soil respiration rates were obtained. Therefore, even small inaccuracies with regard to $Q_{10}$ may result in large errors in the estimation of C dynamic on a global scale [11]. Hence, a better understanding of the processes influencing the temperature sensitivity of SR is required.

In field measurements of SR, the effects of temperature- and non-temperature-related factors on SR are difficult to separate, and their impacts are inevitably confounded. The $Q_{10}$ values determined by in situ SR and soil temperature measurements are known as the apparent $Q_{10}$ due to the confounding effects of biotic and abiotic factors [27]. Some studies have shown that $Q_{10}$ values for HR flux in various biomes may vary widely, from 1 to larger than 12 [25,28,29]. It is suggested that $Q_{10}$ values significantly higher than 2.5 may be caused by the confounded effects of biotic factors such as the substrate supply or vegetation growth [19]. It is also reported that the response of RR to an increase in temperature is higher than the temperature sensitivity of HR, yielding $Q_{10}$ values higher than 4.6 [30]. At the same time, the confounding effects of the soil temperature and moisture on SR significantly reduces the $Q_{10}$ in hot and dry summers in Californian forests [31]. The SR becomes insensitive to the temperature increase because the temperature effect on SR is blocked by the low moisture availability that also results in the limitation of nutrient substrate availability [20,25,31,32]. Therefore, using the apparent $Q_{10}$ value may lead us to overestimate or underestimate the temperature effect on the global SR magnitude and distort the prediction of future soil $CO_2$ emissions in global warming.

The modeling of the annual SR flux and the ecosystem C budget is usually based on the constant $Q_{10}$ value within a year as well. However, some studies have demonstrated that the $Q_{10}$ values of SR rates are not constant throughout the year [16,33,34], because many biotic and abiotic factors, which are responsible for temperature sensitivity of SR, change with season [35]. It is generally suggested that the temperature sensitivity of SR negatively correlates with temperature and tends to reduce with decreasing soil moisture [12,33], as the water deficit can inhibit the microbial activity and root growth [19]. At the same time, the litter input, the quantity and availability of organic substrates, the activity of microbial communities and plant roots are also seasonal fluctuating factors that can affect the temporal (seasonal and interannual) dynamics of the temperature sensitivity of SR [12,36]. Obviously, seasonal changes in abiotic and biotic factors directly affect the dynamics of the two main fluxes in SR—heterotrophic and root respiration. In a 4-year soil warming experiment conducted in a cool-temperate deciduous broadleaved forest in Central Japan, it was illustrated that the warming effect on SR varied seasonally because HR was more sensitive to soil warming in the late-growing season compared to other seasons [37]. The authors highlighted that the projection of the soil carbon cycle in future climate conditions should take seasonal variations in the warming effects on SR into consideration. Therefore, more field studies in various bio-climatic regions are required in order to better understand the mechanisms of the confounded effects of the main abiotic and biotic factors on the temperature sensitivity of SR.

Until now, the interannual dynamics of the temperature sensitivity of SR have not been examined extensively. More attention has been paid to the spatial variations in the temperature sensitivity of soil respiration and its controlling factors. Usually, in situ monitoring of SR in individual ecosystems lasts no more than 3–5 years and is mostly conducted during the growing season; year-round SR measurements in ecosystems with a permanent snow cover are greatly limited [16]. Investigations identifying the factors that drive the interannual variability in $Q_{10}$ at the local and ecosystem levels are practically absent. Such studies are greatly required in order to determine the direction and magnitude of individual ecosystem carbon cycle feedbacks to current climate changes.

Our on-going study aimed to estimate the seasonal and interannual dynamics of the temperature sensitivity of SR based on a long-term 24-year series of measurements in two temperate forest ecosystems in European Russia. The specific objectives of this study were to examine: (1) the correlation between the $Q_{10}$ values and meteorological indexes based on the ratio between the air temperature and precipitation at different time intervals throughout the year, and (2) the impact of the forest type and soil properties on $Q_{10}$ dynamics. The forests studied were formed on soils of different types (Entic Podzol vs. Haplic Luvisol) with contrast textures (sandy vs. loamy) under similar climate conditions. The study sites also differed in terms of the forest type (mixed vs. deciduous), which caused differences in the litter composition, i.e., the substrate quality and availability. Hence, the selected forest ecosystems are characterized by various internal features and identical external (climate) conditions. We hypothesized that the interannual variability of the $Q_{10}$ of SR in temperate forest ecosystems would be determined by the aridity/humidity conditions during the observation period, while the internal soil properties (mainly soil texture) would affect tightness of the relationships between the temperature sensitivity of SR and meteorological indexes.

## 2. Materials and Methods

### 2.1. Study Area and Forest Sites

This study was conducted in the southern part of the Moscow region (Russia), which is part of a boundary between the southern taiga and broad-leaved forest zone (Figure 1). The climate in the region is classified as temperate-continental, with a mean annual temperature of 5.7 °C and an annual precipitation of 640 mm for the period between 1991 and 2020 (current climatic norm). The mean monthly temperatures in July and January for the same period are 18.8 and −7.2 °C, respectively. A permanent snow cover can appear from November to the middle of January and remains until the end of April.

The experiment was carried out in a mature mixed forest (MMF) located in the territory of Prioksko-Terrasny State Reserve (54°50′ N, 37°35′ E) and in a secondary deciduous forest (SDF) adjacent to the Experimental Field Station of the Institute of Physicochemical and Biological Problems of Soil Sciences of the Russian Academy of Sciences (54°20′ N, 37°37′ E). The soils of the studied sites were classified as Entic Podzol (Arenic) in the MMF and Haplic Luvisol (Siltic) in the SDF. The distance between the studied forest sites is about 7.5 km. The soils of the two experimental sites markedly differ in their content of organic carbon (C) and total nitrogen (N), the C to N ratio, and pH value (Table 1). They are also characterized by different values of water holding capacity (WHC), which is mainly due to the differences in soil texture and C content. The carbon stock in the litter of the MMF site is 5.8 ± 0.3 Mg C per ha, which is 2.4 times higher than in the SDF site [38].

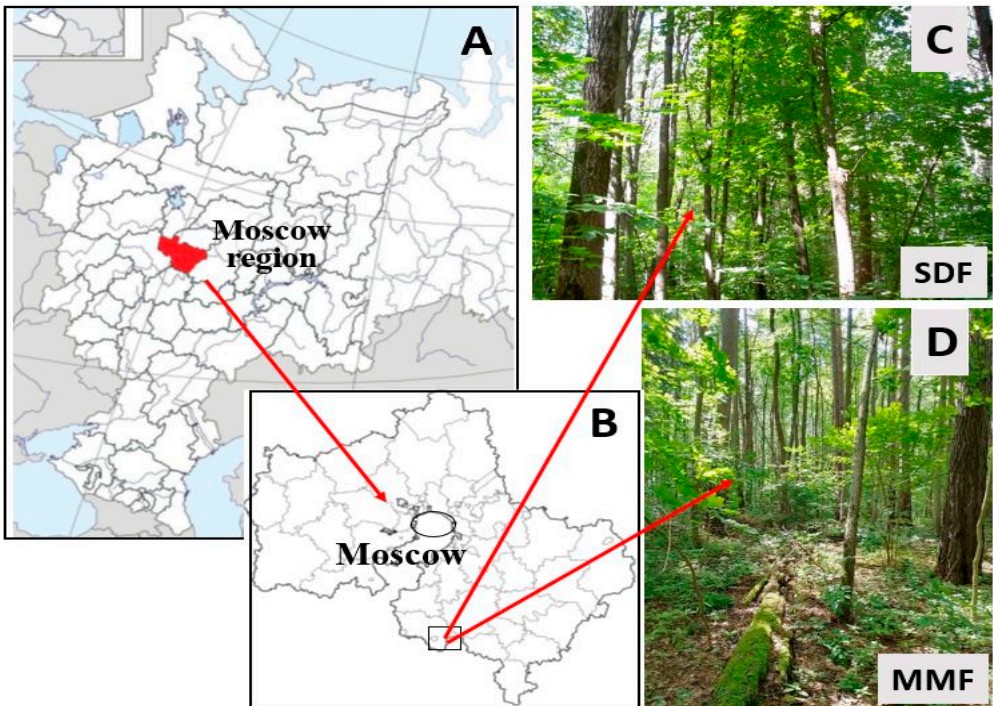

**Figure 1.** Location of the study sites. (**A**)—The European part of Russia; (**B**)—Moscow region; (**C**)—secondary deciduous forest (SDF); (**D**)—mature mixed forest (MMF).

**Table 1.** Basic soil properties (0–10 cm layer) of the forest sites (mean values ± standard error (SE) are shown).

| Site | Soil | Sand:Silt:Clay | WHC, % | C | N | C/N | pH$_{KCl}$ |
|------|------|----------------|--------|---|---|-----|-----------|
| | | | | g/kg of Soil | | | |
| MMF | Entic Podzol Arenic | 11.6:1.0:1.3 | 40.5 ± 2.7 | 12.2 ± 2.8 | 0.96 ± 0.15 | 15.3 | 3.67 ± 0.02 |
| SDF | Haplic Luvisol Siltic | 1.2:2.3:1.0 | 57.5 ± 2.3 | 30.0 ± 1.8 | 2.35 ± 0.10 | 12.8 | 5.56 ± 0.09 |

Note: texture classes: sand: >0.05 mm, silt: 0.002–0.050 mm, clay: <0.002 mm; WHC: water holding capacity.

### 2.2. Soil Respiration Measurements

The SR rate was measured continuously from December 1997 to November 2021 at 7–10-day intervals using a closed chamber method. Measurements were carried out all the year round between 9 and 11 a.m., when the current value of the SR rate was approximately equal to its mean daily value [39,40]. In total, over the 24-year observation period, 1070 and 981 measurements were performed in the MMF and SDF sites, respectively. We conditionally divided the calendar year into two periods: the vegetation period (or snowless period) from May to October and the cold period (mainly with snow cover) from November to April. Gas sampling procedures during the vegetation and the cold periods differed in terms of the size of the chambers used, number of replications, and exposure time [40,41]. Chambers were installed on the soil surface between plants, and the litter was not removed. During the snow-free period, steel (lightproof) cylindrical chambers (10 cm in diameter by 10 cm in height) were used. Chambers were inserted into the soil to a depth of 3–4 cm before the gas sampling. The $CO_2$ concentration in a chamber was determined every 10 min over 30 min or every 15 min over 45 min in the Entic Podzol and Haplic Luvisol, respectively [40,41]. During the cold snowy period (mainly from November to April), 32 × 32 cm steel bases (with a water seal) were inserted permanently to a depth of 20 cm into the soil and 32 × 32 × 15 cm steel boxes were used as a cover. To exclude disturbances from the snow cover, the bases were built up by special sections as required. The increase in the $CO_2$ concentrations in the chambers

was measured over 135 min at 45-min intervals [41]. Gas samples (20 cm$^3$) were collected by a syringe, transported to the laboratory in hermetically sealed vacuumed flasks, and analyzed on the day of sampling using gas chromatographs (Chrom-5, Prague, Czech Republic, or Kristall-2000, Yoshkar-Ola, Russia).

The measured SR flux includes the heterotrophic (mainly microbial) and autotrophic (root) respiration, as well as the non-biological production of $CO_2$. The SR rate was calculated according to the following differential equation, taking into account the suppression of the $CO_2$ evolution due to the decreasing $CO_2$ gradient between the soil air and atmosphere and the inverse diffusion of $CO_2$ into the soil [42]:

$$dC/dt = F_{CO2}/H - Ds/H \times (C - C_0)/Z_{CH} \tag{1}$$

which can be integrated to:

$$C = C_0 + F_{CO2} \times Z_{CH}/Ds \times (1 - \exp(-Ds/(Z_{CH} \times H \times t))), \tag{2}$$

where $F_{CO2}$ is the $CO_2$-C flux, mgC m$^{-2}$ h$^{-1}$; $C_0$ is the initial head-space concentration of $CO_2$-C, mg C m$^{-3}$; C is the head-space concentration of $CO_2$-C, mgC m$^{-3}$, in time t (hour); H is the height of the head-space layer in the chamber, m; Ds is the $CO_2$ diffusion coefficient in the soil; and $Z_{CH}$ is the depth of the lower chamber edge below the soil surface, m.

For the first 20 min during the warm (snow-free) period, we assumed a linear increase in the head-space $CO_2$ concentration in the chamber with time. During the cold season, the net flux of $CO_2$ was negligible within the measurement interval (90 min). Thus, Equations (1) and (2) can be simplified to Equations (3) and (4) [16,40]:

$$dC/dt = F_{CO2}/H \tag{3}$$

or

$$C = C_0 + F_{CO2} \times t/H \tag{4}$$

Simultaneously, the soil moisture (Ms) and soil temperature (Ts) in the upper soil layer (0–5 cm) were determined. Ms (mass %) was determined by the classical gravimetric method, and Ts was measured using a transistorized electrical thermometer TET-2 (Saint-Petersburg, Russia) over 1997–2003. Since November 2003, Ts has been measured by the automatic thermometer Checktemp 1 (Hanna Instruments, Vöhringen, Germany) during the gas sampling procedure and also 6 times per day using the Termochrons iButton (Whitewater, WI, USA).

### 2.3. Data Processing

2.3.1. Calculation of the Meteorological Indexes

The series of long-term meteorological data were provided by the staff of the Station of Background Monitoring (RosHydromet observation Network), located in the territory of Prioksko-Terrasny State Reserve nearby the MMF site. The dataset included the mean daily (and monthly) air temperature (Tair, °C) and the daily (and monthly) level of precipitation (P, mm) for the whole observation period (December 1997 to November 2021). Based on these data, we calculated the following meteorological indexes (MI): (1) mean annual Tair (MAT, °C) and annual sum of precipitation (SP, mm); (2) $ST_{5-8}$ and $ST_{5-9}$ (°C), the sum of the mean monthly Tair from May to August and May to September, respectively; (3) $SP_{5-8}$ and $SP_{5-9}$, the sum of the monthly P from May to August and May to September, respectively; (4) the wetness indexes, $WI_{5-8}$ and $WI_{5-9}$, which present the $lg(SP_{5-8}/ST_{5-8})$ and $lg(SP_{5-9}/ST_{5-9})$, respectively [43]; and (5) the Selyaninov hydrothermal coefficient ($HTC_{6-8}$) over the summer period (June–August), which is the ratio between the sum of precipitation (mm) for the period with a mean daily air temperature above 10 °C to the sum of the temperatures for the same period divided by 10 [16,38]. We believe that all these parameters are the most sensitive predictors of the interannual variability in $Q_{10}$ values, since they cover various time intervals within the growing season, reflecting the differences

in water supply. We also calculated the current climatic norm (ClimNorm) for each MI, which is the mean value of an index over the period between 1991–2020 (Table S1).

2.3.2. Calculation of the Temperature Coefficient $Q_{10}$

The responses of the SR rate to Ts at a 5 cm depth were assessed using the linearized $Q_{10}$ function [44]:

$$Ln(SR) = a + bTs \tag{5}$$

A logarithmic transformation of the SR values yielded a linear function with homoscedastic errors [45]. Then, the parameters of this equation were used to estimate the apparent $Q_{10}$ of the soil respiration, which measures the factor of the increase in the SR rate associated with an increase of 10 °C in the soil temperature:

$$Q_{10} = \exp(10b) \tag{6}$$

We also calculated $SR_{10}$, the rate of soil respiration at 10 °C:

$$SR_{10} = Q_{10} \times \exp(a) \tag{7}$$

The $Q_{10}$ values and $SR_{10}$ were estimated individually and for the whole dataset for: (i) years with different levels of humidity (normal, wet, dry), (ii) the individual calendar seasons of winter (December–February), spring (March–May), summer (June–August) and fall (September–October); and (iii) each of 24 observation years. For every data set, calculations of $Q_{10}$ and $SR_{10}$ were performed for the whole interval of the Ts ($Q*_{10}$ and $SR*_{10}$) and also for the Ts above 1 °C ($Q_{10}$ and $SR_{10}$) to avoid confusing our results with the effects of the SR burst caused by the freezing-thawing processes [46,47]. In this study, we propose keeping the typical symbols $Q_{10}$ and $SR_{10}$ for the values calculated at Ts $\geq 1$ °C, since the majority of studies on the temperature sensitivity of SR are carried out in the positive range of the Ts.

We used the coefficient of determination ($R^2$) and sum of residual squares (SSR) to assess the goodness of fit. All correlations between Ln(SR) and Ts are presented in the Supplementary Materials (Figures S1–S8). Based on the experimental data set, we calculated the observed $SR_{10}$ values (obs-$SR_{10}$ as an average value at Ts = 9.5–10.4 °C) for the whole dataset and individually for the years with different levels of humidity, for the individual calendar seasons, and for each of the 24 observation years.

2.3.3. Statistical Analyses

Before the statistical analyses, all independent variables ($Q*_{10}$, $Q_{10}$, $SR*_{10}$, and $SR_{10}$ values and meteorological indexes) were tested for the normality of distribution and homogeneity of variance (Shapiro-Wilk test). The variability of the meteorological parameters, $Q*_{10}$, $Q_{10}$, $SR*_{10}$, and the $SR_{10}$ values was assessed using the following parameters (Tables S1 and S3): the variation range (VR = Max − Min), oscillation coefficient (Cos) as the ratio between the VR and the mean value (Cos = VR/Mean), and the variation coefficient (CV, %) as the ratio between the standard deviation (STD) and the mean value (CV = STD/Mean × 100). A one-way ANOVA was used to estimate the effect of the soil/forest type on the mean values of $Q*_{10}$, $Q_{10}$, $SR*_{10}$ and the $SR_{10}$ values (*t*-test). Relationships between $Q_{10}$ and the $R_{10}$ values and various MI were explored using the Pearson correlation (F test). The classical metrics of the ANOVA (including the standard deviation, standard error and median, lower and upper quartiles) were also calculated (Tables S1 and S3; Figures 2–4). We also divided the 24-year series of the MI into 4–6 classes (Table S2) and then mean values of $Q_{10}$ for each class were calculated. The Spearman's correlation was used to estimate the relationship between the mean values of $Q_{10}$ and the MI ranks. The paired *t*-test with the adjustment of the Holm–Bonferroni method was applied to assess the statistical significance of the differences between the average values of $Q_{10}$ corresponding to the various classes of MI. Statistical analyses were performed with

Excel MS Office 2013 or with the R software [48], using the $\alpha = 0.05$ level of significance unless stated otherwise.

## 3. Results

### 3.1. Analysis of the Meteorological Indexes from 1998–2021

The meteorological indexes varied significantly over the observation period (Figure 2). The hydrothermal coefficient for the summer season ($HTC_{6-8}$) and the indexes related to precipitation level during the vegetation season ($SP_{5-8}$, $SP_{5-9}$) were the most variable over the 24-year observation period (CV = 29%–44% and Cos > 1.0), whereas the temperature indexes ($ST_{5-8}$, $ST_{5-9}$) for the same period were characterized by the lowest variability, with CV = 6%–7% and Cos < 0.32 (Table S2). Based on the values of the wetness indexes ($HTC_{6-8}$, $WI_{5-8}$ and $WI_{5-9}$), we determined the humidity level for each observation year. If the value of any parameter was above or below its mean value in the observation period by more than 1 STD, the year was referred to as wet or dry, respectively. There were 5 years classified as 'wet', including 1999, 2003, 2006, 2008 and 2020, whereas 7 years were classified as 'dry', including 2002, 2007, 2010, 2011, 2014, 2018 and 2021. The remaining 12 years were referred to as 'normal' in terms of humidity levels, including 1998, 2000, 2001, 2004, 2005, 2009, 2012, 2013, 2015, 2016, 2017 and 2019. In total, the observation period (1998–2021) was slightly warmer and drier than the period for which the climatological norm was assessed (1981–2010).

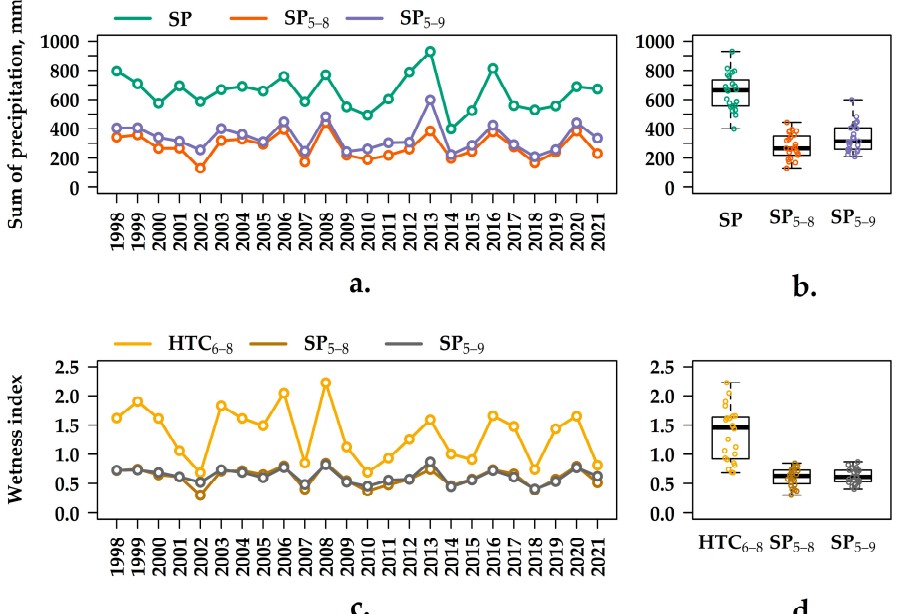

**Figure 2.** Dynamics of the meteorological indexes during the observation period (**a,c**) and their statistical characteristics (**b,d**): the median (bar), lower (Q1) and upper (Q3) quartiles ("boxes"); X1 = Q1 − 1.5 IQR (interquartile range, IQR = Q3 − Q1) and X2 = Q3 − 1.5 IQR ("moustaches"). All data are shown as dots. Meteorological indexes: SP is the annual sum of precipitation (mm); $SP_{5-8}$ and $SP_{5-9}$ are the sum of monthly precipitation from May to August and May to September, respectively; $WI_{5-8}$ and $WI_{5-9}$ are the wetness indexes from May to August and May to September, respectively; $HTC_{6-8}$ is the hydrothermal coefficient over the summer period (June–August).

### 3.2. $Q_{10}$ Values at Various Soil Temperature Intervals and at Different Levels of Humidity

The temperature sensitivity of SR computed for the full range of Ts ($Q^*_{10}$) was 2.47 and 2.26 for the sandy Entic Podzol (MMF) and for the loamy Haplic Luvisol (SDF), respectively (Figure 3, left). For the interval of Ts $\geq 1$ °C, the corresponding values of $Q_{10}$ were lower and amounted to 2.10 and 1.78, respectively (Figure 3, right). The values of $SR^*_{10}$ and $SR_{10}$ in the sandy Entic Podzol, which were estimated according to Equation (7), comprised 1.71 and

1.77 $gCm^{-2}$ $day^{-1}$, respectively. The Loamy Haplic Luvisol was characterized by similar values of SR*$_{10}$ and SR$_{10}$, which amounted to 1.70 and 1.74 $gCm^{-2}$ $day^{-1}$, respectively.

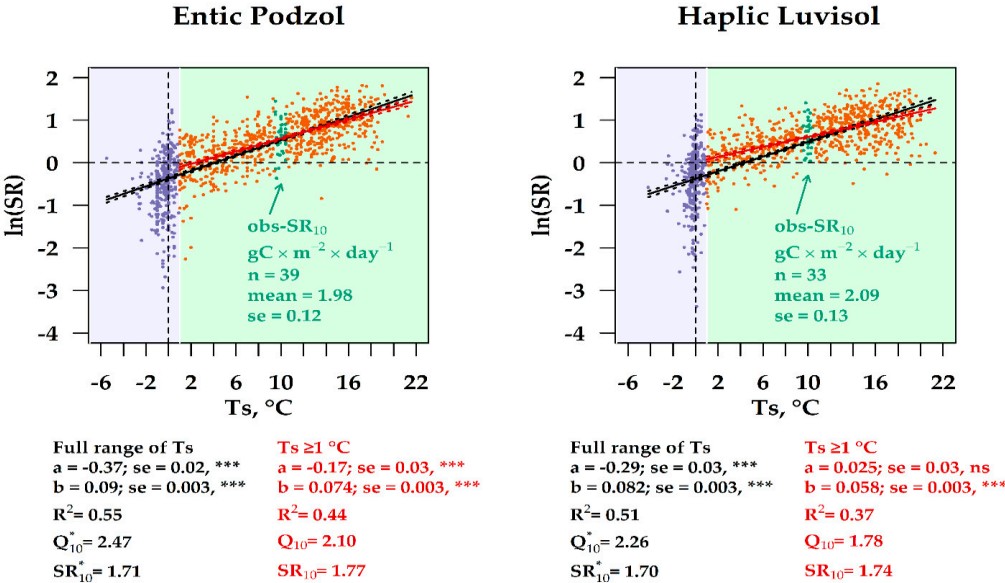

**Figure 3.** Relationship between LnSR and soil temperature, with Ts at 5 cm in the Entic Podzol and Haplic Luvisol for the whole data set (black line) and for Ts >1 °C (red line). SR is the soil respiration rate, $gCm^{-2}$ $day^{-1}$; Ts is the soil temperature at 5 cm; $R^2$ is the coefficient of determination; n is the number of observations; Q*$_{10}$ and Q$_{10}$ are the temperature sensitivity of soil respiration at the full range of Ts (black color) and for Ts $\geq$ 1 °C (red color), respectively. ***—parameters of regression equation (Ln(SR) = a + bTs) are significant at $p < 0.0001$. ns is not significant.

The temperature sensitivities of SR over the years which were normal with regard to humidity were approximately equal for both studied forest soils (Q*$_{10}$ =2.66–2.72), whereas the Q$_{10}$ values for SR at Ts $\geq$ 1 °C in the sandy Entic Podzol were higher than those in the loamy Haplic Luvisol: 2.33 vs. 2.09 (Table 2). Both the Q*$_{10}$ and Q$_{10}$ values for SR in the studied forests in dry years were 1.3–1.4 times lower as compared to the years with normal levels of humidity (Table 2). For the years with high levels of humidity, we also revealed a decrease in the temperature sensitivity of SR in both soils by 11%–18% in comparison to a normal year, except for the Q$_{10}$ values in the Entic Podzol. In the years with normal and high levels of humidity, the Ts at 5 cm was responsible for 42%–61% of the SR variability, whereas in the dry years, only 28%–47% of the SR variation could be explained by the Ts fluctuation at 5 cm.

**Table 2.** Temperature coefficients (Q*$_{10}$ and Q$_{10}$) of soil respiration, SR*$_{10}$, and SR$_{10}$ values for periods of different levels of humidity in the two types of forest.

| Level of Humidity | Q*$_{10}$ | n | $R^2$ | SR*$_{10}$ | Q$_{10}$ | n | $R^2$ | SR$_{10}$ |
|---|---|---|---|---|---|---|---|---|
| **Mixed mature forest, Entic Podzol** | | | | | | | | |
| Normal | 2.72 | 524 | 0.59 | 1.80 | 2.33 | 361 | 0.51 | 1.85 |
| Wet | 2.42 | 225 | 0.61 | 1.86 | 2.32 | 157 | 0.52 | 1.89 |
| Dry | 2.13 | 315 | 0.45 | 1.48 | 1.64 | 207 | 0.29 | 1.57 |
| **Secondary deciduous forest, Haplic Luvisol** | | | | | | | | |
| Normal | 2.66 | 473 | 0.56 | 1.85 | 2.09 | 324 | 0.50 | 1.98 |
| Wet | 2.17 | 211 | 0.54 | 1.89 | 1.80 | 143 | 0.42 | 2.01 |
| Dry | 1.89 | 297 | 0.47 | 1.41 | 1.49 | 204 | 0.28 | 1.54 |

Note: Q*$_{10}$ and Q$_{10}$ are the temperature coefficients of soil respiration at the full range of the soil temperature, Ts, and for Ts > 1 °C, respectively; n is the number of observations; $R^2$ is the coefficient of determination; SR*$_{10}$ and SR$_{10}$ are the rates of soil respiration at 10 °C ($gCm^{-2}$ $day^{-1}$) calculated using Equation (7). All the regression equations are significant ($p < 0.0001$).

The estimated values of SR*$_{10}$ and SR$_{10}$ in both forest ecosystems over the years with normal and higher levels of humidity were similar (Table 2), ranging between 1.80 and 2.01 gCm$^{-2}$ day$^{-1}$. Over the dry years, the values of SR*$_{10}$ and SR$_{10}$ for both forests studied were considerably lower and changed from 1.41 to 1.47 gCm$^{-2}$ day$^{-1}$.

### 3.3. Seasonal Variation of the $Q_{10}$ Values in the Two Types of Forest

The Q*$_{10}$ values for the spring season (March–May) in both studied forests were higher in comparison with the fall season: 2.47–3.13 vs. 1.86–2.33 (Table 2). The temperature sensitivity of SR in the Haplic Luvisol at Ts $\geq$ 1 °C demonstrated a similar tendency, whereas in the Entic Podzol, the Q$_{10}$ value in fall was higher than that in the spring season: 2.14 vs. 1.82. During the spring–fall time, the fluctuations in the Ts at 5 cm were responsible for 26%–47% of the SR variability.

The temperature sensitivity of SR in both forests was lowest during the summer season (June–August). The relationship between SR and Ts at 5 cm for the Haplic Luvisoil was not significant. During the winter season, the temperature sensitivity of SR was the highest, at Q*$_{10}$ = 3.23–6.54, but the Ts fluctuations at 5 °C were responsible only for 7%–8% of the SR variability.

The estimated values of SR*$_{10}$ and SR$_{10}$ in the mixed mature forest (Entic Podzol) for the spring season were similar to those for the fall period and varied between 1.70 and 1.77 gCm$^{-2}$ day$^{-1}$ (Table 3). In the secondary deciduous forest (Haplic Luvisol), the values of SR*$_{10}$ and SR$_{10}$ amounted to 2.00 gCm$^{-2}$ day$^{-1}$ for the spring season and were slightly lower (1.78 gCm$^{-2}$ day$^{-1}$) during the fall period.

**Table 3.** Temperature coefficients (Q*$_{10}$ and Q$_{10}$) of soil respiration, SR*$_{10}$, and SR$_{10}$ values for different calendar seasons in the two types of forest.

| Season | Q*$_{10}$ | n | $R^2$ | SR*$_{10}$ | Q$_{10}$ | n | $R^2$ | SR$_{10}$ |
|---|---|---|---|---|---|---|---|---|
| **Mixed mature forest, Entic Podzol** | | | | | | | | |
| Spring | 2.47 | 272 | 0.34 | 1.76 | 1.82 | 154 | 0.26 | 1.70 |
| Fall | 2.33 | 281 | 0.40 | 1.79 | 2.14 | 259 | 0.34 | 1.77 |
| Winter | 3.23 | 241 | 0.07 | nd | 1.14 | 40 | ns | nd |
| Summer | 1.55 | 276 | 0.06 | nd | 1.55 | 276 | 0.06 | nd |
| **Secondary deciduous forest, Haplic Luvisol** | | | | | | | | |
| Spring | 3.13 | 243 | 0.47 | 2.00 | 2.17 | 152 | 0.43 | 2.00 |
| Fall | 1.86 | 263 | 0.35 | 1.78 | 1.75 | 240 | 0.27 | 1.78 |
| Winter | 6.54 | 216 | 0.08 | nd | 2.14 | 20 | ns | nd |
| Summer | 1.03 | 249 | ns | nd | 1.03 | 249 | ns | nd |

Note: Q*$_{10}$ and Q$_{10}$ are the temperature coefficients of soil respiration at the full range of the soil temperature, Ts, and for Ts $\geq$ 1 °C, respectively; n is the number of observations; $R^2$ is the coefficient of determination; SR*$_{10}$ and SR$_{10}$ are the rates of soil respiration at 10 °C (gCm$^{-2}$ day$^{-1}$) calculated using Equation (7). nd is not determined: the reference SR*$_{10}$ and SR$_{10}$ values for Ts < 10 °C (winter) and for Ts > 10 °C (summer) were not estimated). All the regression equations are significant ($p < 0.0001$) except ns—not significant.

### 3.4. Interannual Variation of the $Q_{10}$ Values in the Forest Ecosystems

The Q*$_{10}$ values ranged from 1.50 to 5.04 in the sandy Entic Podzol and varied between 1.48 and 6.29 in the loamy Haplic Luvisol (Figure 4a) over the 24-year observation period. The interannual variability of the Q*$_{10}$ values (CV) in these soils amounted to 33% and 40%, respectively (Table S3). The coefficients of oscillation of the Q*$_{10}$ values were 1.31 and 2.03 for the Entic Podzol and Haplic Luvisol, respectively. The median value of Q*$_{10}$ in the Entic Podzol was higher than in the Haplic Luvisol by a factor of 1.2, although the difference between the mean values of Q*$_{10}$ was not significant (Figure 4b).

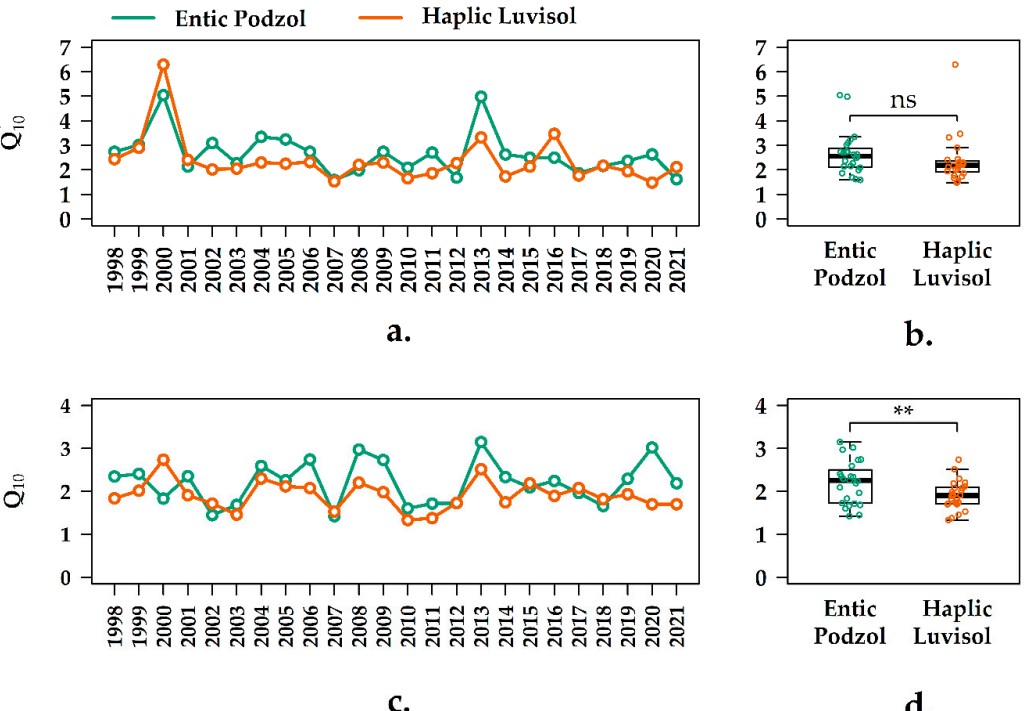

**Figure 4.** Dynamics of Q*$_{10}$ and Q$_{10}$ values in the two forests during the observation period (**a,c**) and their statistical characteristics (**b,d**): the median (bar), lower (Q1) and upper (Q3) quartiles ("boxes"); X1 = Q1–1.5 IQR (interquartile range IQR = Q3–Q1) and X2 = Q3–1.5 IQR ("moustaches"). All data are shown as dots. ns—the difference between mean values is not significant. **—the difference is significant at *p* < 0.01.

The values of Q$_{10}$ for the interval of Ts ≥ 1 °C were considerably lower than the corresponding Q*$_{10}$ values and changed from 1.42 to 3.15 in the sandy Entic Podzol and from 1.33 to 2.74 in the loamy Haplic Luvisol (Figure 4c). The coefficients of the interannual variation and the oscillation coefficients for the Q$_{10}$ values were also lower than the corresponding coefficients for Q*$_{10}$, being 18%–23% (CV) and 0.73–0.78 (Cos) (Table S3). The difference between the mean Q$_{10}$ values in the soils studied was significant at *p* < 0.01 (Figure 4d). The median value of Q$_{10}$ in the Entic Podzol was 1.2 times higher than in the Haplic Luvisol.

*3.5. Interannual Variation of the SR$_{10}$ Values in the Forest Ecosystems*

The values of SR*$_{10}$ ranged from 1.15 to 2.86 gCm$^{-2}$ day$^{-1}$ in the sandy Entic Podzol and changed in the similar interval (1.10–2.65 gCm$^{-2}$ day$^{-1}$) in the loamy Haplic Luvisol over the 24-year observation period (Figure 5a). The interannual coefficients of variation and the oscillation coefficients for the SR*$_{10}$ values in these soils amounted to 22%–23% and 0.88–0.97, respectively (Table S3, Supplementary Materials). The median value of SR*$_{10}$ in the Entic Podzol was similar to that in the Haplic Luvisol. There was no significant difference between the mean values of SR*$_{10}$ in both soils studied (Figure 5b).

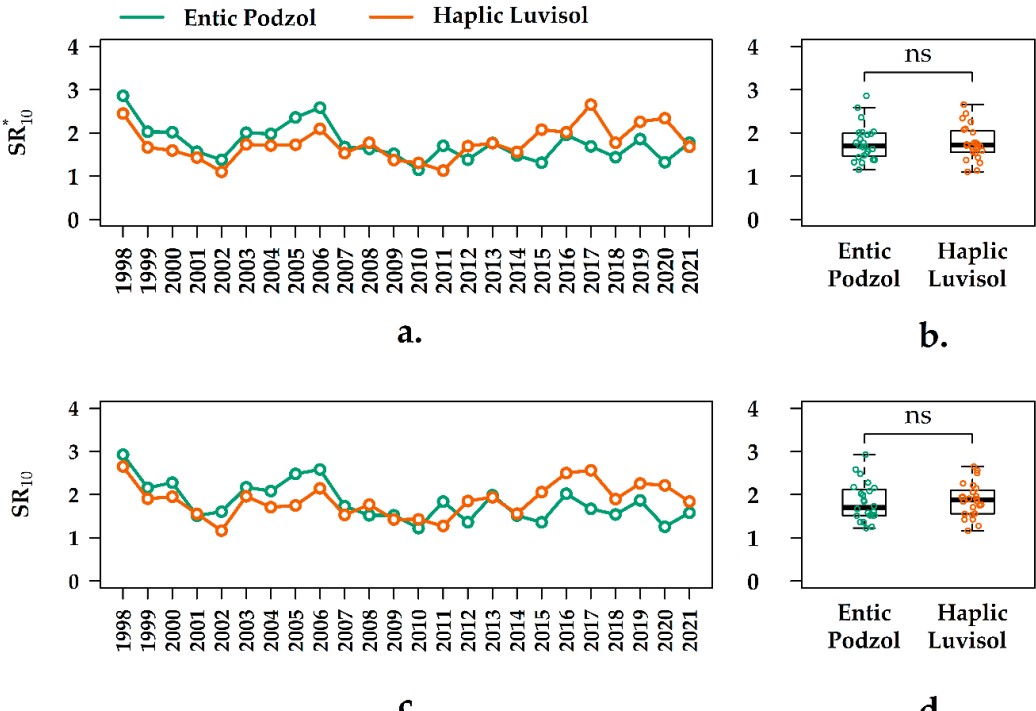

**Figure 5.** Dynamics of SR*$_{10}$ and SR$_{10}$ values (gCm$^{-2}$ day$^{-1}$) in the two forests during the observation period (**a,c**) and their statistical characteristics (**b,d**): the median (bar), lower (Q1) and upper (Q3) quartiles ("boxes"); X1 = Q1 − 1.5 IQR (interquartile range = Q3 − Q1) and X2 = Q3 − 1.5 IQR ("moustaches"). All data are shown as dots. ns—the difference between mean values is not significant.

The values of SR$_{10}$ (for the interval of Ts ≥ 1 °C) for the studied soils changed in ranges which were similar to the corresponding SR*$_{10}$ values: from 1.15 to 2.86 gCm$^{-2}$ day$^{-1}$ in the sandy Entic Podzol and from 1.17 to 2.65 gCm$^{-2}$ day$^{-1}$ in the loamy Haplic Luvisol (Figure 5c). The SR*$_{10}$ and SR$_{10}$ values demonstrated significant negative linear trends during the measurement period (1998–2021), which equated to −0.28 and −0.35 gCm$^{-2}$ day$^{-1}$ per 10 years, respectively (R$^2$ = 0.22–0.31; $p < 0.02$). The coefficients of the interannual variation and the oscillation coefficients for the SR$_{10}$ values were also lower than the corresponding SR*$_{10}$ values and amounted to 21%–24% (CV) and 0.79–0.93 (Cos) (Table S3). The Entic Podzol demonstrated a higher variability in the SR$_{10}$ values in comparison with the loamy Haplic Luvisol. The median values of SR$_{10}$ in the Entic Podzol and Haplic Luvisol amounted to 1.70 and 1.87 gCm$^{-2}$ day$^{-1}$, respectively. There was no significant difference between the mean values of SR$_{10}$ in both soils studied (Figure 5d).

### 3.6. Experimental vs. Observation SR$_{10}$ Values in the Two Forest Ecosystems

The values of obs-SR$_{10}$ changed from 1.08 to 3.21 gCm$^{-2}$ day$^{-1}$ in the sandy Entic Podzol and from 1.26 to 3.28 gCm$^{-2}$ day$^{-1}$ in the loamy Haplic Luvisol (Figure 6a). During the first 8 years of observation (1998–2006), the obs-SR$_{10}$ in the Entic Podzol was higher than in the Haplic Luvisol. This ratio was reversed between 2015 and 2021. There was a significant negative linear trend of obs-SR$_{10}$ in the Entic Podzol during the measurement period (1998–2021), which equated to −0.51 gCm$^{-2}$ day$^{-1}$ per 10 years (R$^2$ = 0.41; $p < 0.001$).

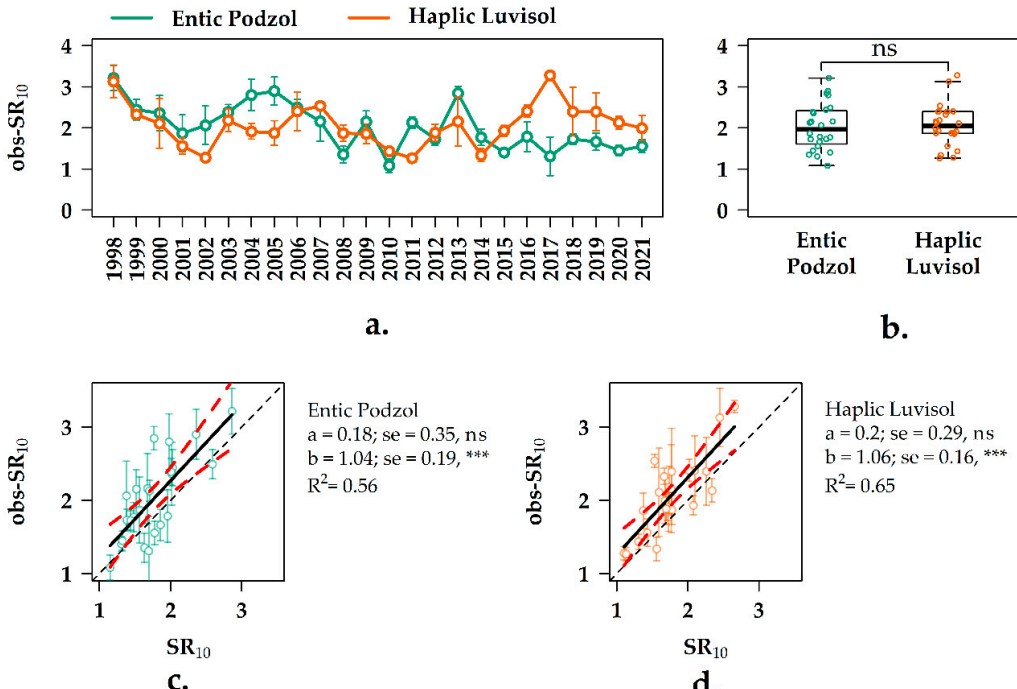

**Figure 6.** Dynamics of obs-SR$_{10}$ values (gCm$^{-2}$ day$^{-1}$) in the two forests during the observation period (**a,c**) and their statistical characteristics (**b**): the median (bar), lower (Q1) and upper (Q3) quartiles ("boxes"); X1 = Q1 − 1.5 IQR (interquartile range = Q3 − Q1) and X2 = Q3 − 1.5 IQR ("moustaches"). All data are shown as dots. ns—the difference between mean values is not significant. Relationship between obs-SR$_{10}$ and SR$_{10}$ for both forest soils (**c,d**). ***—the parameters of regression equation are significant at $p < 0.0001$. ns—not significant. Black solid line is a regression line (significant at $p < 0.001$). Red dotted line corresponds to the standard error, se). Vertical bars are standard errors (±se).

The coefficients of the interannual variation and the oscillation coefficients for obs-SR$_{10}$ in both soils were close: 25%–28% (CV) and 0.97–1.05 (Cos) (Table S3, Supplementary Materials). The median values of obs-SR$_{10}$ in the Entic Podzol and Haplic Luvisol amounted to 1.96 and 2.05 gCm$^{-2}$ day$^{-1}$, respectively. There was no significant difference between the mean values of obs-SR$_{10}$ in both soils studied (Figure 6b). A close relationship ($R^2$ = 0.56–0.65; $p < 0.001$) was observed between the obs-SR$_{10}$ and SR$_{10}$ values throughout the observation period in both forest soils (Figure 6c,d).

*3.7. Relationship between the Q$_{10}$ and SR$_{10}$ Values and Meteorological Indexes over the 24-Year Observation Period*

A negative relationship between the Q$_{10}$ values and some temperature indexes (ST$_{5-8}$ and ST$_{5-9}$) was observed in the loamy Haplic Luvisol (Figure 7). Additionally, the Q$_{10}$ values correlated positively with SP$_{5-8}$. In the Entic Podzol, a positive correlation was found between the Q$_{10}$ values and all precipitation indexes. For both soils, we observed a positive relationship between the temperature sensitivity of SR at Ts ≥ 1 °C and the wetness indexes. The identical coefficients of correlation were usually higher in the sandy Entic Podzol in comparison with the loamy Haplic Luvisol. The meteorological indexes had no significant effect on the Q*$_{10}$ values, except for the positive correlation between the Q*$_{10}$ values and SP$_{5-9}$ in the Entic Podzol.

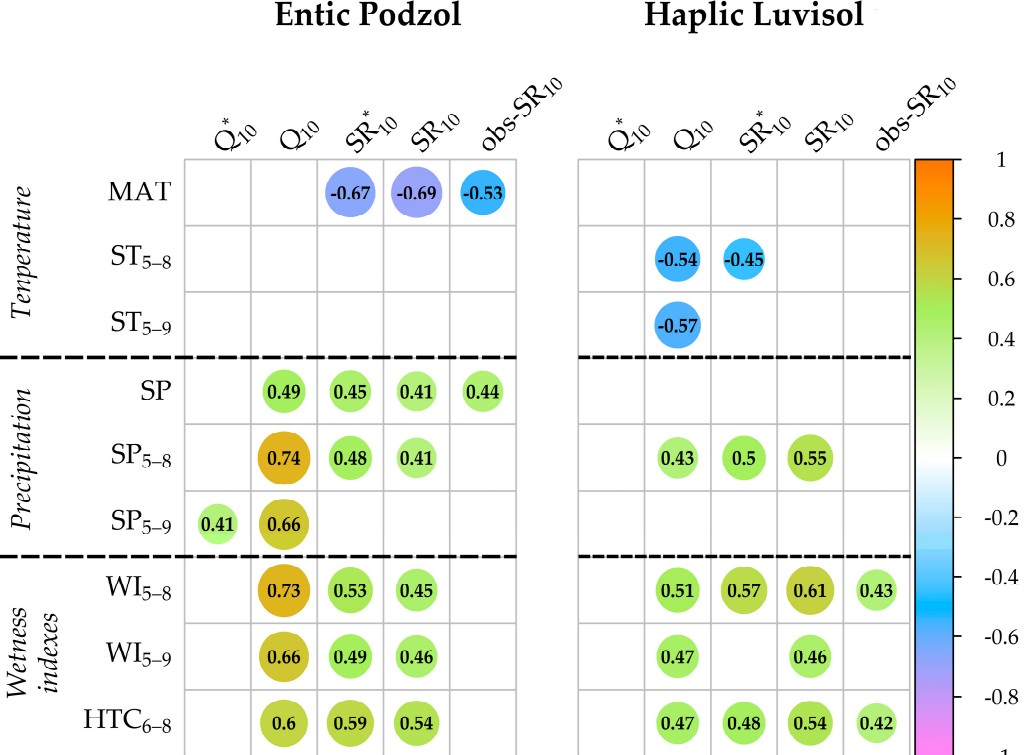

**Figure 7.** Correlations between $Q^*_{10}$, $Q_{10}$, $SR^*_{10}$ and $SR_{10}$ values and the meteorological indexes. Only the significant correlations are shown ($p < 0.05$). $Q^*_{10}$ and $Q_{10}$ are the temperature coefficients of the soil respiration at all ranges of the soil temperature, Ts, and for Ts $\geq$ 1 °C, respectively; $SR^*_{10}$ and $SR_{10}$ are the rates of soil respiration at 10 °C (gCm$^{-2}$ day$^{-1}$) calculated using $Q^*_{10}$ and $Q_{10}$. Obs-$SR_{10}$ is the observation SR value at Ts = 10 °C. Meteorological indexes: SP is the annual sum of precipitation (mm); SP$_{5-8}$ and SP$_{5-9}$ are the sums of monthly precipitation from May to August and May to September, respectively; WI$_{5-8}$ and WI$_{5-9}$ are the wetness indexes from May to August and May to September, respectively; HTC$_{6-8}$ is the hydrothermal coefficient over the summer period (June–August).

Over the whole observation period, the $SR^*_{10}$ and $SR_{10}$ values in the sandy Entic Podzol demonstrated a significant positive correlation with the wetness indexes, SP and SP$_{5-8}$. Additionally, the $SR^*_{10}$ and $SR_{10}$ values were negatively correlated with MAT (Figure 7). In the loamy Haplic Luvisol, the $SR_{10}$ values correlated positively with all wetness indexes and SP$_{5-8}$. The $SR^*_{10}$ values in this soil demonstrated a negative relationship with ST$_{5-8}$ and a positive correlation with SP$_{5-8}$, WI$_{5-8}$ and HTC$_{6-8}$. The obs-$SR_{10}$ value in the Entic Podzol correlated negatively with MAT and positively with SP, whereas the obs-$SR_{10}$ value in the Haplic Luvisol positively correlated with WI$_{5-8}$ and HTC$_{6-8}$.

### 3.8. Changes in the Mean $Q_{10}$ Values for Various Ranks of Meteorological Indexes

The mean values of $Q_{10}$ in the sandy Entic Podzol negatively responded to the growth of the rank values ST$_{5-8}$ and ST$_{5-9}$. At the same time, they demonstrated a positive response to the increase in the rank values SP, SP$_{5-8}$ and SP$_{5-9}$ and all the wetness indexes, which were responsible for 70–99% of the variability in the rank mean $Q_{10}$ values (Figure 8). The lowest $Q_{10}$ values (mean $\pm$ se) were observed at SP$_{5-8}$ $\leq$ 200 mm ($Q_{10}$ = 1.69 $\pm$ 0.17), WI$_{5-8}$ $\leq$ 0.40 ($Q_{10}$ = 1.56 $\pm$ 0.17), WI$_{5-9}$ $\leq$ 0.50 ($Q_{10}$ = 1.76 $\pm$ 0.20) and HTC$_{6-8}$ $\leq$ 0.80 ($Q_{10}$ = 1.57 $\pm$ 0.06). The highest $Q_{10}$ values were observed at SP$_{5-8}$ $\geq$ 381 mm, SP$_{5-9}$ $\geq$ 441 mm ($Q_{10}$ = 2.97 $\pm$ 0.08), WI$_{5-8}$ $\geq$ 0.71 ($Q_{10}$ = 2.86 $\pm$ 0.12), WI$_{5-9}$ $\geq$ 0.81 ($Q_{10}$ = 3.06 $\pm$ 0.09) and HTC$_{6-8}$ $\geq$ 2.01 ($Q_{10}$ = 2.86 $\pm$ 0.12).

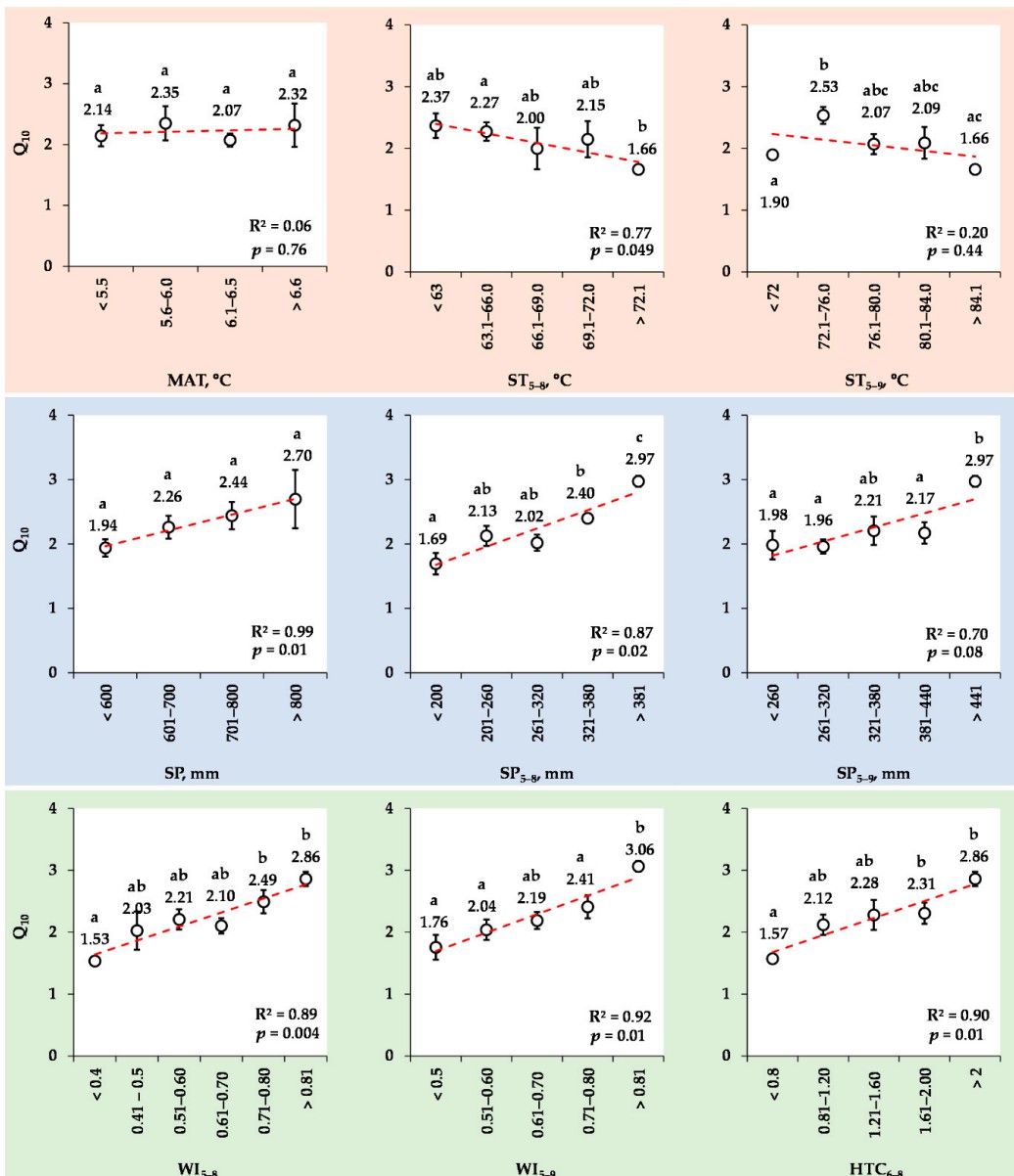

**Figure 8.** Rank correlations between the mean $Q_{10}$ values in the sandy Entic Podzol and meteorological indexes: MAT, °C mean annual Tair; $ST_{5-8}$ and $ST_{5-9}$ (°C) are the sums of the mean monthly Tair from May to August and May to September, respectively; SP is the annual sum of precipitation (mm); $SP_{5-8}$ and $SP_{5-9}$ are the sums of the monthly precipitation from May to August and May to September, respectively; $WI_{5-8}$ and $WI_{5-9}$ are the wetness indexes from May to August and May to September, respectively; $HTC_{6-8}$ is the hydrothermal coefficient over the summer period (June–August). Letters indicate significant differences at $p < 0.05$.

Similar to the sandy Entic Podzol, the mean values of $Q_{10}$ in the loamy Haplic Luvisol also negatively responded to the growth of the rank values $ST_{5-8}$ and $ST_{5-9}$, which were responsible for 73%−81% of their variability (Figure 9). At the same time, the rank mean $Q_{10}$ values demonstrated a positive response to the increase in the rank values SP, $SP_{5-8}$ and $SP_{5-9}$ and all the wetness indexes. However, these positive relationships were less close than they were in the sandy Podzol, explaining 24%−85% of variability in the rank mean $Q_{10}$ values.

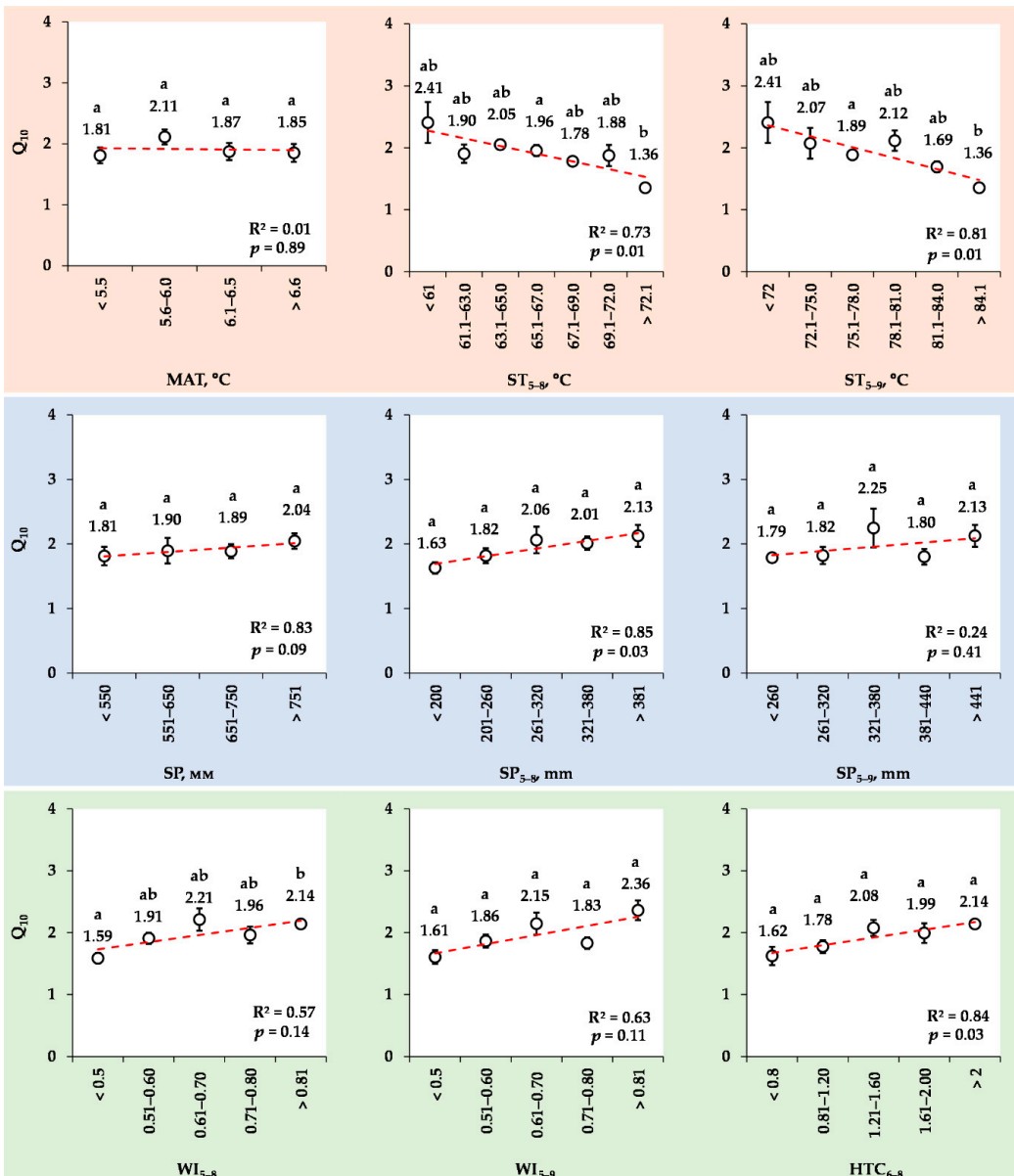

**Figure 9.** Rank correlations between the mean $Q_{10}$ values in the loamy Haplic Luvisol and the meteorological indexes: MAT, °C mean annual Tair; $ST_{5-8}$ and $ST_{5-9}$ (°C) are the sums of the mean monthly Tair from May to August and May to September, respectively; SP is the annual sum of precipitation (mm); $SP_{5-8}$ and $SP_{5-9}$ are the sums of the monthly precipitation from May to August and May to September, respectively; $WI_{5-8}$ and $WI_{5-9}$ are the wetness indexes from May to August and May to September, respectively; $HTC_{6-8}$ is the hydrothermal coefficient over the summer period (June–August). Letters indicate significant differences at $p < 0.05$.

The lowest $Q_{10}$ values were observed at $ST_{5-8} \geq 72.1$ °C, $ST_{5-9} \geq 84.1$ °C ($Q_{10} = 1.36 \pm 0.02$), $SP_{5-8} \leq 200$ mm ($Q_{10} = 1.63 \pm 0.09$), $WI_{5-8} \leq 0.50$ ($Q_{10} = 1.59 \pm 0.08$), $WI_{5-9} \leq 0.50$ ($Q_{10} = 1.61 \pm 0.11$) and $HTC_{6-8} \leq 0.80$ ($Q_{10} = 1.62 \pm 0.15$). The highest $Q_{10}$ values were observed at $ST_{5-8} \leq 61$ °C, $ST_{5-9} \leq 72$ °C ($Q_{10} = 2.41 \pm 0.33$), $SP_{5-8} \geq 381$ mm ($Q_{10} = 2.13 \pm 0.17$), $WI_{5-9} \geq 0.81$ ($Q_{10} = 2.36 \pm 0.16$) and $HTC_{6-8} \geq 2.0$ ($Q_{10} = 2.14 \pm 0.06$).

## 4. Discussion

### 4.1. Impacts of Internal and External Factors on the Temperature Sensitivity of Soil Respiration

The $Q_{10}$ values based on the field measurements of SR depend on the observation period, which may cover the entire year or be limited to the growing season [49]. The

range of soil temperatures for estimating the $Q_{10}$ values can vary markedly, especially in boreal regions, where freezing events are typical in soils. In the soils studied, the Ts at 5 cm varied from −5.5 to 21.6 °C during the observation period (Figure 3), indicating that the soils were regularly subjected to freeze-thaw processes. This remarkable increase in the SR rate due to the soil thawing is well known [47,50–52]; the $Q_{10}$ can reach very high values (10.5–36.6) over the thawing period [46,53,54]. To avoid confusing our results with the effect of the SR burst on the estimations of the SR temperature sensitivity, we calculated the $Q_{10}$ values for the full range of Ts ($Q^*_{10}$) and for Ts above 1 °C ($Q_{10}$). This procedure allowed us to decrease the interannual variability of the $Q_{10}$ values (Figure 3), to identify the key indicators controlling the magnitude of $Q_{10}$ (Figures 7–9), and to correctly compare the temperature sensitivity of the forest soils in various bio-climatic regions where the regular processes of freezing-thawing of the soils are absent.

According to our estimations, the temperature sensitivity of SR in the soils studied in the positive range of Ts was lower by a factor of 1.2 than the corresponding $Q^*_{10}$ values within the full range of Ts, amounting to 1.78 and 2.10 for the Haplic Luvisol and Entic Podzol, respectively (Figure 3). This was in agreement with the results of reports on field studies in Japanese forest soils (40 sites); there, the mean $Q_{10}$ value was 2.12, ranging between 1.30 and 3.17 [55]. The $Q_{10}$ values in China's forests range from 1.10 to 5.18 depending on the climatic and geographical factors, with a mean value of 2.51 [56]. A higher temperature sensitivity of SR was demonstrated in the pine forest of the middle taiga (Komi Republic, Russia), for which the average $Q_{10}$ value was 3.06 over the period between 2014 and 2017 [57]. The temperature sensitivity of soil respiration in a mountain larch forest in in Northeast China was estimated between 5.6 and 7.2 at different depths (0, 5 and 10 cm) within a positive interval of Ts [54]. On a global scale, the area-weighted mean values of $Q_{10}$ in different types of forest vary from 1.50 to 1.75 [11]. Therefore, the temperature sensitivity of SR in the forest ecosystems in European Russia in a temperate-continental climate was slightly higher in comparison with the average worldwide values of $Q_{10}$ for forest soils and lower than the $Q_{10}$ values of SR in the forests of the colder Eurasian regions.

In regard to our results, both the $Q^*_{10}$ and $Q_{10}$ values in sandy the Entic Podzol were higher than in loamy the Haplic Luvisol, whereas the estimated $SR^*_{10}$ and $SR_{10}$ values were identical in both soils (Figure 3). The higher temperature sensitivity of microbial respiration in the sandy Entic Podzol in comparison with the loamy Haplic Luvisol was revealed at an early stage in the short incubation experiments at various levels of soil moisture [58]. Assuming the equal contributions of root respiration to the total SR in both forest soils, the difference in the temperature sensitivity of their SR could be caused by the distinctions in soil texture, which is responsible for water supply in the dry summer period and for the filtration of excess water over the wet seasons. Thus, the respiration of the sandy soils may be more sensitive to water stress, and less suppressed due to the moisture oversaturation, than the respiration of clay soils. Our results demonstrated a remarkable decrease in the $Q^*_{10}$ and $Q_{10}$ values in both soils during the dry years compared to the period with normal levels of humidity (Table 2). For the wet years, in comparison to the normal ones, the temperature sensitivity of SR reduced more considerably in the loamy Luvisol than in the sandy Podzol. The moisture limitation during the dry years also resulted in a considerable decrease in the $SR^*_{10}$ and $SR_{10}$ values, whereas the effect of the high humidity levels was very weak or absent.

The studied forest soils markedly differed in their substrate quality, namely the stock and the composition of the forest litter [38], the content of organic C, the C/N ratio and the pH value (Table 1), as well as the content of microbial C [58]. Clay soils usually contain more recalcitrant (or mineral-associated) organic C than sandy soils, which are more enriched by labile components of soil organic matter (SOM) [59]. It is considered that the quality and the degradability of SOM is an important regulator of $Q_{10}$ [25]. However, up to the present time, it has been actively debated whether the decomposition of labile and recalcitrant SOM pools is equally sensitive to the temperature increase. Some studies demonstrated that recalcitrant and complex SOM compounds are more sensitive to warming than fresh

and labile organic substrates [27,60,61]. Others have reported higher temperature sensitivities for less recalcitrant SOM [62–65], or identical $Q_{10}$ values among various SOM pools [66,67]. Therefore, the temperature sensitivity of SR in the studied forest soils was caused by the confounded effects of internal factors such as soil texture, SOM content and composition, as well as levels of humidity or aridity (external factors). The $SR_{10}$ values were relatively independent from the internal features of the soils studied, but they reacted to the moisture limitation.

*4.2. Seasonal Variability of the Temperature Sensitivity of Soil Respiration*

Today, there is evidence indicating that the temperature sensitivity of SR is not constant throughout the year and changes with the seasons [33,37,54]. In general, $Q_{10}$ values are maximal in the cold period and decline with the temperature increase [11,12,64]. In line with many previous studies, our results indicated a clear seasonal dynamic of $Q^*_{10}$ in both studied soils, with the highest values (3.23 and 6.54) in the winter period and the lowest ones (1.03 and 1.55) over the summer season (Table 3). Therefore, our results fully support the seasonal plasticity hypothesis [68], which suggests that the microbial activity and enzyme structure adapts evolutionarily or acclimates physiologically in response to seasonal changes in temperature and thermal constraints. Often, these processes are characterized by higher temperature sensitivities in colder environments and their decline with increasing temperatures.

However, the pattern of seasonal dynamics in $Q_{10}$ values may vary due to the large spatial heterogeneity in climatic and hydrothermal conditions, as well as in the SOM quality and decomposability [25,54]. For instance, in the larch mountain forests in Northeast China, soil respiration rates were most sensitive to temperature changes in the early stage of near-surface soil thawing ($Q_{10}$ = 11.5–36.6) and in the middle of the growing period ($Q_{10}$ = 6.5–40.2), reflecting the higher responsiveness of SR to changes in hydrometeorology and ground freeze-thaw processes [54]. A strong effect of the climatic zone on the seasonal dynamics of the temperature sensitivity of SR was revealed in an incubation experiment on soils located along the latitude gradient of eastern China [69]. In identical natural mixed forests, the spring and autumn temperature sensitivities of SR increased toward the southern region, but the $Q_{10}$ values were similar across the latitude gradient in summer. It was suggested that the summer $Q_{10}$ values were closely associated with the dominance of microbial r-strategy features, characterized by high copiotroph/oligotroph and labile/recalcitrant C degradation gene ratios. The spring $Q_{10}$ values were independent of the microbial community composition and functions, increasing with the reducing C availability from north to south. At the same time, the autumn $Q_{10}$ values were driven by the K-selected microbial communities, which may be ascribed to the priming effects mediated by fresh plant litter [69]. Therefore, the microbial community composition and exo-enzyme production vary considerably in response to the seasonal changes in substrate and nutrient availability, which might be one of the reasons for the seasonal dynamic in the $Q_{10}$ values [20,37,70].

Davidson et al. [19] reported that SR always exhibited higher $Q_{10}$ values in spring than in fall, perhaps because of springtime root growth and a hysteresis phenomenon based on soil temperatures, which were measured at a fixed depth, that resulted in varying levels of $CO_2$ production over the spring and fall seasons. In our study, both the $Q^*_{10}$ and $Q_{10}$ values exhibited the same ratio only for the SR of the clay Haplic Luvisol in the secondary deciduous forest, whereas in the sandy Entic Podzol in the mixed mature forest, the $Q^*_{10}$ and $Q_{10}$ values over spring and fall time were almost equal (Table 3). This difference in spring–fall $Q_{10}$ pattern can be explained by various dynamics in the available substrate concentrations in the soils studied. Due to the faster turnover of forest litter in the SF site compared to the MMF one (0.9 vs. 2.8 year) [71], there is a deficit of easily decomposable SOM in the clay Haplic Luvisol during the late summer and fall periods, whereas the sandy Entic Podzol is characterized by a nearly constant concentration of available C throughout the year. The estimated $SR^*_{10}$ and $SR_{10}$ values were more or less identical in both soils

during spring and fall seasons (Table 3), indicating their independence from the seasonal dynamics of the external and internal factors.

Thus, the seasonal dynamic in the temperature sensitivity of SR in the forest soils was caused by changes in both external (or abiotic) factors, such as hydrometeorology and freeze-thaw processes, and internal (or biotic) ones, including the dynamics in the litter input and its turnover, supplied substrate quality and availability, and the ability and strategy of the microbial community in decomposing SOM. For a better understanding of the seasonal dynamic in the $Q_{10}$ values in the studied forest soils, year-round measurements of SR should be accompanied by the monitoring of the monthly dynamics in the substrate and nutrient availability for the decomposer microorganisms.

*4.3. Interannual Variability of the Temperature Sensitivity of SR in the Forest Ecosystems*

In many ecosystem carbon cycle models, it is generally suggested that the temperature sensitivity of SR is constant from year to year [12,72]. Understanding the factors controlling the interannual variability in the temperature sensitivity of SR in soils under temperate-continental climate conditions is crucial for improving our ability to assess climate feedback in the future. Some published studies have clearly demonstrated a rather wide range of the $Q_{10}$ values from year to year in individual ecosystems. For instance, Osipov et al. [57] reported that the temperature sensitivity of SR over the growing season in the pine forest of the middle taiga (Komi Republic, Russia) ranged from 2.2 to 3.7 over the period between 2014 and 2017. In the sub-alpine forests (Eastern Qinghai-Tibet Plateau, China) over the period of 2004–2006, the $Q_{10}$ values of the growing season varied between 3.96 and 6.82 [33]. In the Tibetan alpine grassland, the temperature sensitivity of SR varied between 2.89–5.59 and 2.29–2.89 during the growing and non-growing periods, respectively, over a 4-year observation period (2009–2012) [32].

In regard to the results of the 24-year observation presented in this study, the temperature sensitivity of SR demonstrated a high temporal interannual variability (Figure 4; Table S3), with coefficients of variation (CV) = 33%–40% for $Q^{*}_{10}$ (range 1.33–2.74) and 18%–23% for $Q_{10}$ (range 1.48–6.29). The lowest $Q_{10}$ values for SR in both studied soils (1.4–1.7) were registered over the dry years (e.g., 2002, 2007 and 2010), which were characterized by the lowest wetness indexes and $HTC_{6-8}$ values (Figure 2, Table S1). Therefore, our data support the first hypothesis, that the moisture availability is a key factor influencing the temperature sensitivity of soil respiration in forest ecosystems in temperate-continental climates. In general, the increase in the temperature accelerates the microbial respiration due to increases in both the activity of extracellular enzymes that degrade the complex polymeric components of SOM and the rates of microbial uptake of soluble substrates [20,70]. Moisture limitation can suppress the microbial activity, regardless of the temperature, which should result in a reduction in the temperature response of SR to temperature changes [27,32,34]. The temperature sensitivity SR responds to moisture availability because it influences the diffusivity of soluble organic substrates, which is low at low water content, and oxygen diffusivity, which is low at a high water content [11]. The low diffusivity of any soluble substrates at a low water content or the oxygen limitation at a high water content reduces the respiration of soil microbiota [19]. Hence, there is a confounding effect of the main abiotic and biotic factors on the temperature sensitivity of SR, as estimated by the field studies.

The identification of the drivers controlling the temperature sensitivity of SR is usually based on the meta-analysis of field or incubation studies which are carried out along the latitudinal or climatic gradients [11,55]. On the whole, there is a negative relationship of the $Q_{10}$ values with the temperature and a positive correlation with the soil moisture level. Based on the unique continuous 24-year measurements of SR, we have shown that the main controlling factor for the temperature sensitivity of SR in the sandy Entic Podzol at Ts $\geq$ 1 °C is the moisture supply during the growing season. There were significant correlations between the $Q_{10}$ values and wetness/hydrothermal indexes over various periods of the growing season, as well as the precipitation levels over the year and for the

late-spring–summer season (Figure 7). In the clay Haplic Luvisol, similar correlations were less pronounced, but additionally, a negative relationship between the $Q_{10}$ values and the sum of the average temperatures for different periods of the growing season was identified. Therefore, the set of meteorological indexes controlling the interannual variability of the temperature sensitivity of forest soils in temperate-continental climates depends on the soil texture, which is responsible for the water supply during the growing season. The reference values of the SR rate at 10 °C (both $SR*_{10}$ and $SR_{10}$) demonstrated a relationships similar to that of $Q_{10}$, indicating the close relationship between the controlling factors and the mechanisms of their interannual variability. There were no close correlations between the temperature sensitivity of SR at the full range of Ts ($Q*_{10}$ values) and the studied meteorological indexes. The estimates of the rank correlations between $Q_{10}$ and the meteorological indices confirmed the relationships described above and allowed us to determine the average values of $Q_{10}$ for the various ranges of the meteorological indexes (Figures 8 and 9). These $Q_{10}$ values could be used to estimate the SR rates in forest ecosystems of boreal regions, taking into account the weather conditions of each year studied.

Therefore, we can conclude that our hypothesis was fully confirmed. Our results clearly demonstrated that the water supply, which was determined by the aridity/humidity conditions over the growing season, was a key driver of the interannual variability of the $Q_{10}$ values in the temperate forest ecosystems, while the soil texture affected the nature of the relationship between the temperature sensitivity of SR and the meteorological indexes. Considering the clear tendency towards climate aridization in the region studied [16] and in many other regions of the world [73], we can predict a reducing temperature sensitivity of soil respiration and assume a rising carbon sink in the boreal and temperate forest ecosystems.

### 4.4. Research Perspectives in Light of the Current Climate Changes

In the future, the simulation of SR rates employing various $Q_{10}$ values may be a promising method for improving the accuracy of annual SR flux estimates used for simulating soil carbon sources and sinks in the studied area. To better understand the temporal variation of the $Q_{10}$ values, the monthly or seasonal monitoring of the substrate and nutrient availability for soil, as well as the composition and strategy of the microbial community, is also needed.

As an extremely sensitive component of the carbon cycle, the SR can reflect the current state of the climate in a region and respond to long-lasting changes in the hydrothermal regime [74]. The last ICCP report [75] informed us that the ability of natural ecosystems to provide carbon storage and sequestration is increasingly influenced by heat, wildfire, droughts and other impacts. In the future, with increasing global warming, more regions in the world will be affected by increases in agricultural, soil and ecological droughts [75–77]. For instance, an increase in the frequency and duration of droughts in the territory of the Russian Federation will be observed not only in regions with lower levels of precipitation, but also in regions where there are distinct trends of increasing total humidification levels [76]. Significant correlations between the temperature sensitivity of SR and wetness/hydrothermal indexes over different periods of the growing season, using the average values of $Q_{10}$ for various ranges of the meteorological indexes which were observed in the present study, will allow us to improve the prognosis of SR fluxes on the local and regional scales.

Regional climate models (RCMs) are an important source of climate information at the regional scale [75,78]. The key reason for enhancing their use as a source of regional information is their high-scale resolution [78,79]. In particular, simulations at a kilometer-scale resolution add value to the prognosis of sub-daily precipitation extremes, soil-moisture–precipitation feedback [75], and trends in seasonal air temperatures [78,79]. This is especially important for regions with complex landscapes [78]. Therefore, the coupled application of RCMs and the findings of the present study could contribute greatly to the assessment of soil respiration fluxes in changing climates.

## 5. Conclusions

In this study, we conducted a synthesis of the 24-year year-round measurements of soil respiration in two forest ecosystems with temperate-continental climates and estimated, for the first time, the interannual variability of the $Q_{10}$ values. Additionally, the drivers of seasonal and interannual heterogeneity of the $Q_{10}$ values were determined. Our results indicated that, when the year-round SR dynamic is investigated, it is reasonable to estimate $Q_{10}$ functions in the positive range of the soil temperature (Ts $\geq$ 1 °C) in order to avoid confusing the results with the effect of freezing-thawing on the soil respiration rate and $Q_{10}$ values. We clearly demonstrated that the main controlling factor of the temperature sensitivity of SR in the sandy Entic Podzol at Ts $\geq$ 1 °C was the moisture availability during the growing season. There were significant correlations between the $Q_{10}$ values and wetness/hydrothermal indexes over various periods of the growing season, as well as precipitation levels over the year and for the late spring–summer season. In the clay Haplic Luvisol, similar correlations were less pronounced, but additionally, a negative relationship between the $Q_{10}$ values and the sum of the average temperatures for different periods of the growing season was found. The estimates of the rank correlations between $Q_{10}$ and the meteorological indexes allowed us to determine the average values of $Q_{10}$ for various ranges of more sensitive meteorological parameters. These $Q_{10}$ values could be used to estimate the SR rates in forest ecosystems of boreal regions, taking into account the weather conditions of each year studied. Under the current trend of climate aridity intensification, we may predict a reduction in the temperature sensitivity of soil respiration and assume a rise in the carbon sink in boreal and temperate forest ecosystems. In the future, a simulation of the SR rate employing various $Q_{10}$ values can provide a method to improve the accuracy of annual SR flux estimates and simulate the carbon balance in the forest ecosystems.

**Supplementary Materials:** The following supporting information can be downloaded at: https://www.mdpi.com/article/10.3390/f13091374/s1, Table S1: Statistical characteristics for meteorological indexes for the observation period ($n$ = 24); Table S2: Ranks of meteorological indexes for calculation of the Spearmen's correlation coefficient; Table S3: Statistical characteristics for $Q_{10}$ and $SR_{10}$ values for different temperature intervals ($n$ = 24); Figure S1: Correlation between the logarithmic function of the soil respiration rate (Ln(SR)) and soil temperatures (Ts) at 5 cm at the full range of Ts in sandy Entic Podzol in 1998–2009; Figure S2: Correlation between the logarithmic function of the soil respiration rate (Ln(SR)) and soil temperatures at 5 cm (Ts) at the full range of Ts in sandy Entic Podzol in 2010–2021; Figure S3: Correlation between the logarithmic function of the soil respiration rate (Ln(SR)) and soil temperatures (Ts) at 5 cm at the full range of Ts in clay Haplic Luvisol in 1998–2009; Figure S4: Correlation between the logarithmic function of the soil respiration rate (Ln(SR)) and soil temperatures (Ts) at 5 cm at the full range of Ts in clay Haplic Luvisol in 2010−2021; Figure S5: Correlation between the logarithmic function of the soil respiration rate (Ln(SR)) and soil temperatures (Ts) at 5 cm at Ts $\geq$ 1 °C in sandy Entic Podzol in 1998–2009; Figure S6: Correlation between the logarithmic function of the soil respiration rate (Ln(SR)) and soil temperatures (Ts) at 5 cm at Ts $\geq$ 1 °C in sandy Entic Podzol in 2010−2021; Figure S7: Correlation between the logarithmic function of the soil respiration rate (Ln(SR)) and soil temperatures (Ts) at 5 cm at Ts $\geq$ 1 °C in clay Haplic Luvisol in 1998–2009; Figure S8: Correlation between the logarithmic function of the soil respiration rate (Ln(SR)) and soil temperatures (Ts) at 5 cm at Ts $\geq$ 1 °C in clay Haplic Luvisol in 2010–2021.

**Author Contributions:** Conceptualization, I.K.; methodology, V.L.d.G.; investigation, D.S., T.M., V.L.d.G., D.K. and V.Z.; writing—original draft preparation, I.K.; writing—review and editing, I.K.; visualization—D.K. and V.L.d.G. All authors have read and agreed to the published version of the manuscript.

**Funding:** This research was funded by Russian Scientific Foundation (grant number 22-24-00691).

**Institutional Review Board Statement:** Not applicable.

**Informed Consent Statement:** Not applicable.

**Data Availability Statement:** The data presented in this study are available upon request from the corresponding author.

**Acknowledgments:** We are grateful to Vera Ableeva (Atmospheric Monitoring Station, Prioksko-Terrasny State Biospheres Reserve, Danki, Moscow region, Russia) for providing the meteorological data set for 1973–2021. We are very thankful to Anna López de Guereñu (University of Potsdam, Germany) for the grammatical and stylistic editing of the English. We also thank the two anonymous reviewers for their constructive comments, which helped us to improve the manuscript.

**Conflicts of Interest:** The authors declare no conflict of interest.

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
