# Peer review of "Temperature Sensitivity of Soil Respiration in Two Temperate Forest Ecosystems: The Synthesis of a 24-Year Continuous Observation"

_forests, doi:10.3390/f13091374_

Round 1

Reviewer 1 Report

This study deals with the temperature sensitivity of soil respiration in two temperate forest ecosystems: synthesis of a 24-year continuous observation.

It is interesting and well structured. Also, it is easy to follow and the reader has a clear flow of the representation of the results. However, some improvements are necessary and more details are needed to add so as considered accepted.

The climate change and the effects on natural ecosystem must be highlighted so as the reader better understand the necessity of such analysis and need for further research. To that end some outputs of the latest IPCC report must be added (IPCC, 2021). Also mention the RCMs, the most modern tool for future climate projection. Their spatial resolution highly effect models performances especially in areas with complex terrain such as forest ecosystems as found in a recent study about temperature (Stefanidis, 2021).

IPCC. Climate Change 2021: The physical science basis. In Contribution of Working Group, I to the Sixth Assessment Report of the Intergovernmental Panel on Climate Change; Cambridge University Press: Cambridge, UK, 2021.

In the last paragraph of the introduction the novelty points of the current approach must be highlighted in comparison with other similar research to the best author’s knowledge. State the research gap answered from this research.

Line 110. Why did not update these values with more recent data until 2020 for example?

Line 208 Justify why some classical statistical metrics does not use.

Line 223. While previous the authors refer to a period of 1981-2010? Keep a constant period through all the article.

Line 618-621. This text remains from the template “Authors should discuss the results and how they can be interpreted from the perspective of previous studies and of the working hypotheses. The findings and their implications should be discussed in the broadest context possible. Future research directions may also be highlighted.

This is an interesting article and highly important for the readers of the journal. My suggestion in the aforementioned parts I believe will strengthen the sound of the research.

Author Response

August, 15, 2022

Dear Reviewer#1,

We are very thankful to you for the thoughtful reviewing of our manuscript, your valuable suggestions to improve our MS. Thanks a lot for your positive evaluation of our study.

We’ve considered all of your remarks and comments in the revised MS. Following your recommendation, we checked carefully English spelling and stylistics. Below you can find explanations for each one of your comments:

It is interesting and well structured. Also, it is easy to follow and the reader has a clear flow of the representation of the results. However, some improvements are necessary and more details are needed to add so as considered accepted.

The climate change and the effects on natural ecosystem must be highlighted so as the reader better understand the necessity of such analysis and need for further research. To that end some outputs of the latest IPCC report must be added (IPCC, 2021). Also mention the RCMs, the most modern tool for future climate projection. Their spatial resolution highly effect models performances especially in areas with complex terrain such as forest ecosystems as found in a recent study about temperature (Stefanidis, 2021).

IPCC. Climate Change 2021: The physical science basis. In Contribution of Working Group, I to the Sixth Assessment Report of the Intergovernmental Panel on Climate Change; Cambridge University Press: Cambridge, UK, 2021.

Thank you for your valuable suggestions. We included an additional section to the Discussion part:

4.4. Research perspectives in the light of current climate changes.

For the future, the simulation of SR rate with employing various Q10 values may be rather promising to improve the accuracy of annual SR flux estimates for simulating soil carbon source and sink in the studied area. To better understand the temporal variation of Q10 values, monthly or seasonal monitoring of substrate and nutrient availability for soil as well as composition and strategy of microbial community is also needed.

Being an extremely sensitive component of the carbon cycle, the SR can reflect the current state of climate in the region and respond to long-lasting changes in the hydrothermal regime [74]. The last ICCP report [75] informs that the ability of natural ecosystems to provide carbon storage and sequestration is increasingly influenced by heat, wildfire, droughts, and other impacts. In the future, with increasing global warming, more regions in the world will be affected by increases in agricultural, soil, and ecological droughts [75, 76,77]. For instance, an increase in the frequency and duration of droughts on the territory of Russian Federation will be observed not only in regions with a lower amount of precipitation, but also in regions where there are distinct trends of in-creasing of total humidification [76]. Significant correlations between temperature sensitivity of SR and wetness/hydrothermal indexes over different periods of growing season and using of the average values of Q10 for various ranges of meteorological indexes, which were observed in the present study, will allow us to improve the prognosis of SR fluxes on local and regional scales.

Regional climate models (RCMs) are an important source of climate information at the regional scale [75, 78]. The key reason for increasing of their utility as a source of regional information is their high scale resolution [78, 79]. In particular, simulations at a kilometer-scale resolution add value of prognosis of sub-daily precipitation extremes, soil-moisture–precipitation feedback [75], and trends in seasonal air temperature [78,79]. This is especially important for regions with complex landscapes [78]. Therefore, the coupled application of RCMs and findings of the present study could contribute greatly to the assessment of soil respiration fluxes under changing climate.

The extra references were also included to the references list:

  1. Lopes de Gerenyu, V.O.; Kurganova, I.N.; Khoroshaev, D.A. The Effect of Contrasting Moistening Regimes on CO2 Emission from the Gray Forest Soil under a Grass Vegetation and Bare Fallow. Eurasian Soil Science 2018, 51, 1200–1213, doi:10.1134/s1064229318100034.
  2. IPCC Climate Change 2021: The Physical Science Basis. Contribution of Working Group I to the Sixth Assessment Report of the Intergovernmental Panel on Climate Change 2021, In Press.
  3. Roshydromet Second Assessment Report on Climate Change and its Consequences in the Russian Federation (General Summary); Roshydromet: Moscow, 2014; ISBN 978-5-904206-13-0.
  4. Stefanidis, S.; Alexandridis, V. Precipitation and Potential Evapotranspiration Temporal Variability and Their Relationship in Two Forest Ecosystems in Greece. Hydrology 2021, 8, 160, doi:10.3390/hydrology8040160.
  5. Stefanidis, S.P. Ability of Different Spatial Resolution Regional Climate Model to Simulate Air Temperature in a Forest Ecosystem of Central Greece. Journal of Environmental Protection and Ecology 2021, 22, 1488–1495.
  6. Stefanidis, S.; Dafis, S.; Stathis, D. Evaluation of Regional Climate Models (RCMs) Performance in Simulating Seasonal Precipitation over Mountainous Central Pindus (Greece). Water 2020, 12, 2750, doi:10.3390/w12102750.

In the last paragraph of the introduction the novelty points of the current approach must be highlighted in comparison with other similar research to the best author’s knowledge. State the research gap answered from this research.

Thanks a lot! We included the required information to the introduction:

‘Until now, the interannual dynamics in temperature sensitivity of SR have not been examined extensively. More attention has been paid to the spatial variation of temperature sensitivity of soil respiration and its controlling factors. Usually, in situ monitoring of SR in individual ecosystems lasts not more than 3-5 years and is mostly conducted during the growing season; year-round SR measurements in ecosystems with a permanent snow cover are strongly limited [16]. Investigations identifying the factors that drive the interannual variability of Q10 at the local and ecosystem levels are practically absent. Such studies are strongly needed to determine the direction and magnitude of in-dividual ecosystem carbon cycle feedback to current climate changes.’

Line 110. Why did not update these values with more recent data until 2020 for example?

In the revised version of MS, we presented the climatic parameters for 1991-2020:

‘The climate in the region is classified as temperate continental with a mean annual temperature of 5.7°C and an annual precipitation of 640 mm for the period between 1991 and 2020. The mean monthly temperatures of July and January for the same period are 18.8 and -7.2°C, respectively.’

Line 208 Justify why some classical statistical metrics does not use.

It is not very clear what the reviewer understands by classical statistical metrics. In the Table 1S and 3S (se, please Supplementary materials), we show the full list of ANOVA metrics. Figures 2-4 demonstrate median, lower and upper quartiles.

We added explanation in the text (subsection Statistical analyses)

‘The classical parameters of ANOVA (including the standard deviation and standard error, median, lower and upper quartiles) were also calculated (Tables 1S and 3S; Figures 2 − 4).’

Line 223. While previous the authors refer to a period of 1981-2010? Keep a constant period through all the article.

Thanks! Here we estimate the meteorological indexes only for observation period which lasts from 1998 to 2021

Line 618-621. This text remains from the template “Authors should discuss the results and how they can be interpreted from the perspective of previous studies and of the working hypotheses. The findings and their implications should be discussed in the broadest context possible. Future research directions may also be highlighted.

Thank you very much for your careful reading of the manuscript!

This is an interesting article and highly important for the readers of the journal. My suggestion in the aforementioned parts I believe will strengthen the sound of the research.

We are very thankful to the anonymous reviewer for the positive response on our study and valuable comments which allowed us to improve our MS.

Prof. Irina Kurganova (on behalf of co-authors)

Reviewer 2 Report

The presented study brings important results from the long term measurements on temporal variability of temperature sensitivity of soil respiration and its relationship to meteorological parameters. As temperature is one of the key factors of soil respiration and is often used for its modelling, it is necessary to precisely estimate the temperature sensitivity and its possible variations in time. The manuscript is written comprehensively, the topic is well presented in the Introduction. The results are presented quite clearly. It was just a little difficult to follow because of a high number of parameters. I have only a few comments which should be considered by the authors.

One key word more than in the instructions

The authors use Q10 as a general term for temperature sensitivity of SR and for the sensitivity calculated for Ts>1°C. This is quite confusing escpecially in discussion. Please consider distinguishing these two terms using some label. And similar comment to SR10

L69-71 the authors stress here the effect of drought on Q10 and soil respiration. It would be proper if they briefly explain the reason (substrate availability limitation)

L87 before, the authors divided the SR into heterotrophic and autotrophic (root). Here, there is heterotrophic and root-rhizosphere. The authors should keep a uniform terminology and. Moreover, rhizosphere respiration includes also microbes, therefore, a part of the heterotrophic respiration

L95 on: The authors should explain why they chose the two forest, or why is the comparison of the selected forests interesting

L101 add “be” after will

L139-140 Please add more information about chamber size and the time of its closure

L144-145 Please specify what sensors were used for determination of soil temperature and moisture

L172-178 The authors should reconsider if all the used meteorological indexes are necessary for the analyses, or to justify them. E.g. indexes for May to August (5-8) and May to September (5-9). There is the difference only one month. Is it relevant? Or why the authors used HTC only for the summer months)

Fig 3: the authors should explain also all the different colours in the graphs, and SR10

L271: I believe that ranging would be better word instead of changing (generally through the manuscript)

Table 3 Do the authors have any specific reason why the seasons are not in the order? (spring-summer-fall-winter). It is quite confusing. The authors should also explain why SR was calculated only for spring and fall

L293 the authors stated that the relationship between SR and Ts was non-significant for both sites, “ns” is stated only for Haplic Luvisol in Table 3

Author Response

August, 15, 2022

Dear Reviewer #2,

We are very grateful to you for the thoughtful reviewing of our manuscript, your valuable suggestions to improve our paper.

We’ve considered all of your remarks and comments (please, see the revised MS with corrections), Below you can find explanations for each one of your comments:

Comments and Suggestions for Authors

The presented study brings important results from the long term measurements on temporal variability of temperature sensitivity of soil respiration and its relationship to meteorological parameters. As temperature is one of the key factors of soil respiration and is often used for its modelling, it is necessary to precisely estimate the temperature sensitivity and its possible variations in time. The manuscript is written comprehensively, the topic is well presented in the Introduction. The results are presented quite clearly. It was just a little difficult to follow because of a high number of parameters. I have only a few comments which should be considered by the authors. 

One key word more than in the instructions

Thanks for your accuracy! We cancelled 1 extra key word -  temperate continental climate;

The authors use Q10 as a general term for temperature sensitivity of SR and for the sensitivity calculated for Ts>1°C. This is quite confusing especially in discussion. Please consider distinguishing these two terms using some label. And similar comment to SR10

We use the symbols Q10 and SR10 in introduction and discussion as the most accepted parameters  in literature. Since most studies of the temperature sensitivity of soil respiration are carried out in the positive range of soil temperatures, we decided that it’s more logical to use these designations for Q10 calculated precisely in this temperature range. So, we add the sign * for Q10, which is calculated for the entire rank of soil T. We agree that there is a confusion with the designations, but using more symbols for these parameters will only add to the confusion. We have added a short explanation to the Methods section regarding the use of notation for Q10 and SR 10:

In this study, we propose to keep the typical symbols Q10 and SR10 for the values calculated at Ts ≥ 1°C, since the majority of studies on the temperature sensitivity of SR are carried out in the positive range of Ts.’

L69-71 the authors stress here the effect of drought on Q10 and soil respiration. It would be proper if they briefly explain the reason (substrate availability limitation)

Thanks for your comment. You are right. We rephrased the sentence:

The SR became insensitive to the temperature increase because the temperature effect on SR is blocked by the low moisture availability that results in the limitation of nutrient substrate availability as well [20,25,31,32].’

 L87 before, the authors divided the SR into heterotrophic and autotrophic (root). Here, there is heterotrophic and root-rhizosphere. The authors should keep a uniform terminology and. Moreover, rhizosphere respiration includes also microbes, therefore, a part of the heterotrophic respiration

Thanks for your comments. We agree that the terminology over the MS should be uniform. So, we corrected this sentence considering your remark. In the revised version of the MS we used root instead of root-rhizosphere

L95 on: The authors should explain why they chose the two forest, or why is the comparison of the selected forests interesting

Thank you very much for your valuable comment. We included the information that you recommended to the end of the introduction (before our hypotheses). It allowed to make our speculations more clear.

The forests studied were formed on the soils of different types (Entic Podzol vs Haplic Luvisol) with contrast texture (sandy vs loamy) under similar climate conditions. The study sites also differed in forest type (mixed vs deciduous) that caused differences in the litter composition, i.e. substrate quality and availability. Hence, the selected forest ecosystems are characterized by various internal features and identical external (climate) conditions.’

L101 add “be” after will

Thanks, done

 L139-140 Please add more information about chamber size and the time of its closure

We included the required information:

‘If shortly, during the snow-free period, steel (lightproof) cylindrical chambers (10 cm in diameter by 10 cm in height) were used. Chambers were inserted into the soil to a depth of 3−4 cm before the gas sampling. The CO2 concentration in a chamber was determined during 30 or 45 min every 10 or 15 min in Entic Podzol and Haplic Luvisol, respectively [40,41]. During the cold snowy period (mainly from November to April), 32×32 cm steel bases (with a water seal) were inserted permanently to a depth of 20 cm into the soil and 32×32×15 cm steel boxes were used as a cover. To exclude disturbances from the snow cover, the bases were built up by special sections as required. The increase of CO2 concentrations in the chamber was measured during 135 min with 45-min intervals [41].’

L144-145 Please specify what sensors were used for determination of soil temperature and moisture

 We included the required information:

‘Ms (mass %) was determined by classical gravimetric method; Ts was measured by transistorized electrical thermometer TET-2 (Russia) during 1998 – 2003. Since November 2003, Ts has been measured by automatic thermometer Checktemp 1 (Hanna Instruments, Germany) during the gas sampling procedure and also 6 times per day by Termochrons iButton (USA).’

L172-178 The authors should reconsider if all the used meteorological indexes are necessary for the analyses, or to justify them. E.g. indexes for May to August (5-8) and May to September (5-9). There is the difference only one month. Is it relevant? Or why the authors used HTC only for the summer months)

Water supply during the late spring (May) and summer periods, which we characterize as the amount of precipitation during the May-August (P5-8), is a key parameter for the development of vegetation and has a strong influence on CO2 emission flux from soils, which includes root respiration as one of the most important components. The period from May to September corresponds to the growing season in the studied region; it is the main contributor to the annual soil respiration flux. Therefore, we included the sum of average monthly temperatures (ST5-8 and ST5-9) and precipitation amounts (SP5-8 and SP5-9) as well as the lg of their ratio (WI5-8 and WI5-9) to the list of main meteorological indices which should be considered in the analyses. Besides, our previous studies (e.g. Kurganova et al, 2017 [38]; 2020 [15]) demonstrated the clear negative of trends for summer hydrothermal coefficient НТС6-8 during the SR measurement period and the close relationship between annual CO2 fluxes and НТС6-8. Thus, we believe that all these parameters should be the most sensitive predictors of interannual variability of Q10 (SR10) values. In the present study, we have demonstrated that some indices work better in Luvisols than in Phaeozems and vice versa. In the future, the list of meteorological parameters can be expanded by April and October to cover the entire snowless period.

We’ve included the short explanation the text:

We believe that all these parameters should be the most sensitive predictors of interannual variability of Q10 values, since they cover various time intervals within the growing season reflecting their differences in water supply.’

Fig 3: the authors should explain also all the different colours in the graphs, and SR10

Thanks. We’ve included the required short explanation (in brackets) to the title of Figure 3.

L271: I believe that ranging would be better word instead of changing (generally through the manuscript)

Thanks! We made 2 corrections in the MS text.

Table 3 Do the authors have any specific reason why the seasons are not in the order? (spring-summer-fall-winter). It is quite confusing. The authors should also explain why SR was calculated only for spring and fall/

Spring and fall are characterized by similar intervals of Ts variation in comparison with summer and winter. The correlations between SR and Ts over these periods are tight and significant. This is the reason why we compared spring and fall and located them together in the Table 3. Winter and summer seasons are the most contrast periods with Ts << 10°C in winter and mostly > 10°C in summer. The correlations between SR and Ts over these periods are weak and sometimes insignificant. These are the reasons why we did not estimate the SR10 values for winter and summer periods.

Now, we use nd (not determined) instead of '-' in table 3. We believe, it's not correct to estimate the reference values of SR10 (at 10°C) for Ts  << 10°C (winter) and at Ts > 10°C (summer).

A short explanation was included to the Note for table 3:

‘nd is not determined (we did not estimate the reference values of SR*10 and SR10 for Ts << 10°C (winter) and for Ts > 10°C (summer).’   

 L293 the authors stated that the relationship between SR and Ts was non-significant for both sites, “ns” is stated only for Haplic Luvisol in Table 3

 Thanks a lot for your careful reading of the MS. We have made this clarification:

‘… the relationship between SR and Ts at 5 cm for Haplic Luvisol was non-significant.

Prof. Irina Kurganova (on behalf of co-authors)

Round 2

Reviewer 1 Report

The article can now be accepted